# PRL-2 phosphatase is required for vascular morphogenesis and angiogenic signaling

Mathilde Poulet [1,2], Jacinthe Sirois[2], Kevin Boyé [1], Noriko Uetani[2], Serge Hardy[2], Thomas Daubon [1], Alexandre Dubrac [3,4], Michel L. Tremblay [2✉] & Andreas Bikfalvi [1✉]

Protein tyrosine phosphatases are essential modulators of angiogenesis and have been identified as novel therapeutic targets in cancer and anti-angiogenesis. The roles of atypical Phosphatase of Regenerative Liver (PRL) phosphatases in this context remain poorly understood. Here, we investigate the biological function of PRL phosphatases in developmental angiogenesis in the postnatal mouse retina and in cell culture. We show that endothelial cells in the retina express PRL-2 encoded by the *Ptp4a2* gene, and that inducible endothelial and global *Ptp4a2* mutant mice exhibit defective retinal vascular outgrowth, arteriovenous differentiation, and sprouting angiogenesis. Mechanistically, *PTP4A2* deletion limits angiogenesis by inhibiting endothelial cell migration and the VEGF-A, DLL-4/NOTCH-1 signaling pathway. This study reveals the importance of PRL-2 as a modulator of vascular development.

[1] Univ. Bordeaux, INSERM, LAMC (Laboratoire de l'Angiogenèse et du Microenvironment des Cancers), U1029, F-33600 Pessac, France. [2] McGill Goodman Cancer Research Center, McGill University, Montréal, QC, Canada. [3] Centre de Recherche, St. Justine, Montréal, QC, Canada. [4] Université de Montréal, Montréal, QC, Canada. ✉email: Michel.tremblay@mcgill.ca; andreas.bikfalvi@u-bordeaux.fr

Angiogenesis, the formation of new blood vessels from preexisting ones, is a fundamental developmental process and is involved in different physiological and pathological processes, including tissue repair, cancer, ischemia, stroke, or Alzheimer disease[1–7]. Many pro-angiogenic or antiangiogenic factors have been identified and their mechanisms of action studied during the past decade. Among them, the vascular endothelial growth factor (VEGF) family has important roles in the angiogenic process. The first and most studied member is VEGF-A, which has a predominant role in sprout formation[8,9]. Indeed, the loss of a single allele of the *Vegfa* gene in mice is sufficient to cause an embryonic lethality at E8.5 due to a severely defective vasculature[8,9]. Deletion of *Vegfr2* gene, the main VEGF receptor in endothelial cells, also induces embryonic lethality at the same embryonic stage with a similar phenotype[10,11].

It has been shown that NOTCH-1/DLL-4 signaling plays a crucial role in vascular development and is involved in the tip–stalk cell crosstalk. Namely, *Dll4*[+/−] mice exhibit a strong retinal phenotype with a delay of the formation of the vascular plexus and hyper-sprouting at the retinal borders[12,13]. Furthermore, the activation of NOTCH-1 directs tip-derived endothelial cells into developing arteries[13]. Thus, DLL-4/NOTCH-1 signaling represents an important molecular link between sprouting angiogenesis and artery formation.

Tyrosine phosphorylation levels are governed by a balanced action between protein tyrosine kinases and protein tyrosine phosphatases (PTPs)[14]. Members of the PTP family have been found to play a plethora of cellular and physiological functions, such as in the regulation of transcription, cell proliferation and migration, embryogenesis, neural plasticity, immunity, and more[15–24]. Aberrant positive or negative phosphatase activities have similarly been implicated in the pathogenesis of many diseases ranging from cancer to metabolic diseases and immune deficiencies[25–34]. Although the roles of tyrosine kinase receptors, such as VEGFR-2, in angiogenesis are well studied, little is known about the function of PTPs in these processes. Only few PTPs modulating angiogenesis have been identified so far. Among these, some act by direct dephosphorylation of the key receptor VEGFR-2, while others directly influence signaling in vascular cells[35–39].

The PRL phosphatases are intracellular phosphatases that are composed of three members, PRL-1, PRL-2, and PRL-3 (gene name mouse/human: *Ptp4a1*/PTP4A1, *Ptp4a2*/PTP4A2, and *Ptp4a3*/PTP4A3) sharing ~75% amino acid sequence identity[40]. They have been classified into the subgroup of VH-like PTPs with putative dual-specificity for phosphotyrosine and phosphoserine[40–42]. PRLs exhibit an extremely low enzymatic activity against PTP synthetic substrates in vitro[43]. This is explained in part by the lack of an important hydroxyl group provided by the serine/threonine amino acid residues, which is present in the catalytic region of most PTPs[44,45]. As a consequence, no common mechanism of action and no substrate has been clearly demonstrated yet, even if various-associated proteins have been reported for each of the three PRLs[46]. Among the PRL family members, PRL-3 has been shown to be expressed in embryonic blood vessels and to be involved in angiogenic processes in tumors[47–49]. Despite the homology with the other PRL members, PRL-3's mechanism of action seems to be very different[50,51]. Furthermore, nothing is known about the role of PRL-2 in the vascular development.

In this article, we investigated the biological relevance and in vivo function of PRL-2 in postnatal vascular development, and provide clear evidence that PRL-2 act as a regulator of developmental angiogenesis.

## Results

**Expression of PRL-2 in the mouse retina.** We searched for *Ptp4a1*, *Ptp4a2*, and *Ptp4a3* expression in published single-cell transcriptomic data. We first analyzed the Tabula Muris public transcriptomics dataset from single cells isolated by fluorescence-activated cell sorting (FACS) of the adult mouse brain, which like the retina is part of the central nervous system (CNS) (http://tabula-muris.ds.czbiohub.org/; ref. [52]). Violin plots showed that *Ptp4a1* was poorly expressed in endothelial cells (mean of 0.58), *Ptp4a3* had an intermediate expression level (mean of 2.94), and *Ptp4a2* had the highest expression (mean of 4.13; Supplementary Fig. 1a–c). In addition, *Ptp4a2* expression in endothelial cells was higher than in pericytes or in any other cell type, while *Ptp4a3* expression was highest in pericytes (Supplementary Fig. 1a–d).

Analysis of another single-cell transcriptome of adult brain endothelial cells (http://betsholtzlab.org/VascularSingleCells/database.html; refs. [53,54]) allowed not only the distinction of gene expression between different cell types (mural cells, fibroblasts, and astrocytes), but also between different types of brain endothelial cells (venous, arterial, and capillary endothelial cells). In this dataset, *Ptp4a1* was expressed at lower levels than *Ptp4a2* and *Ptp4a3* in endothelial cells, while fibroblasts and oligodendrocytes expressed high *Ptp4a1* levels (Supplementary Fig. 1e). *Ptp4a2* was slightly more enriched than *Ptp4a3* in some endothelial cell clusters (EC1 and vEC). *Ptp4a3* was expressed at highest levels in mural cells (vascular smooth muscle and pericytes; Supplementary Fig. 1e). We then evaluated *Ptp4a1*, *Ptp4a2*, and *Ptp4a3* expression in the developing CNS using the Trans-omics Resource Database (VECTRDB; https://markfsabbagh.shinyapps.io/vectrdb/; ref. [55]), which contains scRNA-seq data from FACS-purified GFP-positive endothelial cells isolated from a postnatal day 7 (P7) *Tie2*-GFP mouse brain, to assess *Ptp4a1*, *Ptp4a2*, and *Ptp4a3* expression in various brain endothelial cell subtypes (arterial, venous, tip, capillary, and mitotic endothelial cells). *Ptp4a2* and *Ptp4a3* were expressed at higher levels than *Ptp4a1* in all of these subtypes. *Ptp4a2* was also enriched in capillary, venous, and mitotic endothelial cells (Supplementary Fig. 1f).

Online bulk RNA-sequencing data from P6 to P21 mouse retinal endothelial cells[56] revealed developmental regulation of *Ptp4a2* expression, with higher levels at early time points (P6–P15) and a decrease at P21 (ref. [56]; Supplementary Fig. 2a). By contrast, *Ptp4a1* was expressed at low levels throughout retina development, and *Ptp4a3* expression peaked at P21.

In support of the contention that among the three *Ptp4a* genes, *Ptp4a2* plays an important role in the developing retina vasculature, qPCR analysis of whole retina mRNA extracts demonstrated that *Ptp4a2*, but not *Ptp4a1*, was increased between P5 and P21 (Supplementary Fig. 2b).

We finally examined PRL-1 and PRL-2 expression in several endothelial cell types isolated from human umbilical veins (HUVEC) and arteries (HUAEC), from brain vessels (human cerebral microvascular endothelial cells; hCMEC/D3), as well as from microvascular endothelial cells (HMVEC; Supplementary Fig. 1c). Western blot analysis showed that the different endothelial cells mainly expressed PRL-2, but not PRL-1.

Taken together, the data showed that in addition to PRL-3, which was already identified as an angiogenesis regulator, *Ptp4a2*/PRL-2 was widely expressed in endothelial cells of the nervous system, including the retina, prompting us to study the function of this phosphatase in more detail.

**Endothelial-specific inducible *Ptp4a2* deletion inhibits retinal angiogenesis.** To test endothelial PRL-2 function, we generated *Ptp4a2*[fl/fl] mice and crossed them with *Cdh5*CreERT2 mice, an

extensively validated Cre driver strain that allows tamoxifen (TAM)-inducible, endothelial-specific deletion driven by the VE-cadherin promoter[57]. TAM was injected from P0–P2 and mice were sacrificed at P6 (Fig. 1a). The resulting offspring (*Ptp4a2*[fl/fl] *Cdh5*CreERT2, here after *Ptp4a2*[fl/fl]iEKO) exhibited similar body weight compared to TAM-injected Cre-negative littermate controls (*Ptp4a2*[fl/fl]; Fig. 1b). To determine the efficacy of endothelial *Ptp4a2* deletion, lung endothelial cells were isolated from *Ptp4a2*[fl/fl]iEKO and *Ptp4a2*[fl/fl] mice, and *Ptp4a2* expression was assessed by qPCR (Fig. 1c). *Ptp4a2* expression was reduced in *Ptp4a2*[fl/fl]iEKO mice in comparison to littermate controls ($n = 6$ mice for each condition), while neither *Ptp4a1* nor platelet/endothelial cell adhesion molecule-1 (PECAM-1) expression was altered (Fig. 1c).

Analysis of P6 isolectin B4 (IB4)-stained retinal whole mounts showed an important reduction of vascular development in *Ptp4a2*[fl/fl]iEKO retinas when compared to controls (Fig. 1d). Quantifications revealed a 40% decrease of vascular outgrowth and a decrease in vascular density in *Ptp4a2*[fl/fl]iEKO retinas compared to controls (*Ptp4a2*[fl/fl]iEKO $n = 27$, *Ptp4a2*[fl/fl] $n = 25$; Fig. 1d, e). This effect was already seen in the heterozygote mice (*Ptp4a2*[fl/wt]iEKO), in which vascular outgrowth was inhibited by 20% in comparison to control (Supplementary Fig. 3a, b). Interestingly, at the angiogenic front of developing retinal vessels, vascular sprout number, as well as filopodial protrusions per sprout were increased in the *Ptp4a2*[fl/fl]iEKO mice compared to control littermates (Fig. 1g, h, i). This highlights mis-regulated endothelial tip cell guidance and migration.

We reasoned that the reduction in vascular density could be due to a decrease in endothelial cell proliferation or to an increase in vascular regression. Staining with an antibody against the endothelial-specific nuclear ERG1/2/3 transcription factor revealed reduced endothelial cell numbers in P6 *Ptp4a2*[fl/fl]iEKO retinas ($n = 6$ animals per group; Fig. 2a, b)[58]. However, the percentage of endothelial cells labeled with the proliferation marker KI67 was similar between *Ptp4a2*[fl/fl]iEKO retinas and controls, indicating that endothelial cell proliferation was not affected by loss of PRL-2 function (Fig. 2c, d). In order to investigate vascular stability, we labeled retinas with IB4 and an antibody recognizing the basement membrane protein collagen IV (Coll IV), and quantified IB4-negative, Coll IV-positive empty basement membrane sleeves that are left behind when vessels regress. *Ptp4a2*[fl/fl]iEKO retina presented more positive Coll IV sleeves compared to control littermates, indicating a decrease in vessel stability upon loss of endothelial PRL-2 function (Fig. 2e, f).

**Vascular phenotype in global *Ptp4a2* knockout mice.** We next investigated whether global *Ptp4a2*[−/−] mice also exhibited defective vascular morphogenesis. Western blot analysis of mouse retina lysates from wild-type, heterozygote, and *Ptp4a2*[−/−] mice demonstrated loss of PRL-2 protein in *Ptp4a2*[−/−] mice, while PRL-1 remains expressed (Fig. 3a). Analysis of *Ptp4a2*[−/−] P6 neonates, by IB4 staining, revealed retinal vascular developmental defects, including a decreased vascular outgrowth of ~40% compared to wild-type littermate controls (Fig. 3b, c). Since *Ptp4a2*[−/−] mice are known to exhibit lower body weight compared to WT mice[59] and body weight influences retina vascularization, we also performed our analysis only in mice of the same body weight, which confirmed the defect on retinal vascular coverage in *Ptp4a2*[−/−] mice (Supplementary Fig. 4a, b). Interestingly, and in contrast to endothelial *Ptp4a2* mutants, higher magnification images of the retinal vascularization in *Ptp4a2*[−/−] mice showed an increase in vascular density, especially at the vascular front, which is a hallmark of inadequate and uncontrolled angiogenesis (Fig. 3d, e). The number of vessel branching points was also increased in *Ptp4a2*[−/−] mice by >20% when compared to wild-type littermates (Fig. 3d, f). This phenomenon occurs especially in areas surrounding veins. Quantification of the number of tip cells showed that *Ptp4a2*[−/−] mice retinas exhibited an increase in the number of sprouts, when compared to wild-type littermates (Fig. 3g, h). *Ptp4a2*[+/−] mice showed an intermediate phenotype, which indicates a PRL-2 dose-dependent regulation of the vascular development (Supplementary Fig. 4c–e).

Vascular outgrowth defects in *Ptp4a2*[−/−] mice persisted at least until P9 (Fig. 3i, j). Consequently, the vascular front of the *Ptp4a2*[−/−] P9 neonates did not reach the edge of the retina compared to *Ptp4a2* wild type (wt) mice. This further led to a decrease in vascularization of the retina deep plexus at P9 (Fig. 3i, k).

Overall, the vascular phenotype in global *Ptp4a2*[−/−] mouse was similar to the phenotype in the *Ptp4a2*[fl/fl]iEKO mouse in terms of vascular outgrowth and vascular sprouts. However, differences were seen in vascular density, which suggests that *Ptp4a2* signaling in surrounding non-endothelial cells contributes to its effects on vascular development.

**Arterial and venous patterning defects in *Ptp4a2*[fl/fl]iEKO and *Ptp4a2*[−/−] mice.** In addition to sprouting angiogenesis, both *Ptp4a2*[fl/fl]iEKO and *Ptp4a2*[−/−] mice also exhibited a patterning defect of retinal arteries and veins. Indeed, *Ptp4a2*[fl/fl]iEKO (Fig. 4a) and *Ptp4a2*[−/−] (Fig. 4b) mice had a lower number of main retina arterioles and venules compared to littermate control mice (Fig. 4c–f). This was confirmed by co-staining of IB4 with α-SMA, a specific marker of smooth muscle cells that surround arterial vessels during vascular development. In *Ptp4a2*[−/−] mice at P21, vascular outgrowth and density were similar to control, but the arterial and venous defect was still maintained (Fig. 4g, h). This highlights a role of PRL-2 in patterning and maintenance of arteries and veins.

**PRL-2 regulates endothelial cell migration and sprouting angiogenesis.** We further investigated PRL-2 function in vitro using HUVEC cells, a primary and non-immortalized model of angiogenesis. Downregulation of *PTP4A2* mRNA expression of ~90% was achieved by *PTP4A2* siRNA treatment, without affecting *PTP4A1* or *PTP4A3* mRNA (Fig. 5a). PRL-2 protein expression was decreased by 80% and a slight increase of 20% in PRL-1, which is recognized by the same antibody, was seen in western blots after *PTP4A2* knockdown (Fig. 5b, c). We have also investigated EC shape and cell–cell junction morphology via VE-cadherin staining, after *PTP4A2* knockdown. Cells maintain their original area and circularity, and cell junctions are preserved (Supplementary Fig. 5a–c). Because we observed a delay of vascular coverage in *Ptp4a2*[fl/fl]iEKO and *Ptp4a2*[−/−] mice (Figs. 1e and 3c), we performed scratch wound assays on HUVEC monolayers stimulated with angiogenic growth factors VEGF-A or FGF-2. An inhibition of ~50% in wound closure 16 h after scratch was observed after *PTP4A2* knockdown in response to VEGF-A, but not to FGF-2 (Fig. 5d, e and Supplementary Fig. 6a).

To determine if this effect was due to a decrease in cell proliferation, we performed an EdU incorporation assay on HUVEC cells cultured in endothelial cell growth medium, which was supplemented with VEGF-A. No difference was seen in *PTP4A2* siRNA-treated cells compared to siRNA control cells (Fig. 5f, g). This demonstrates that *PTP4A2* knockdown had no effect on endothelial cell proliferation and is consistent with the KI67 in vivo staining (Fig. 2c).

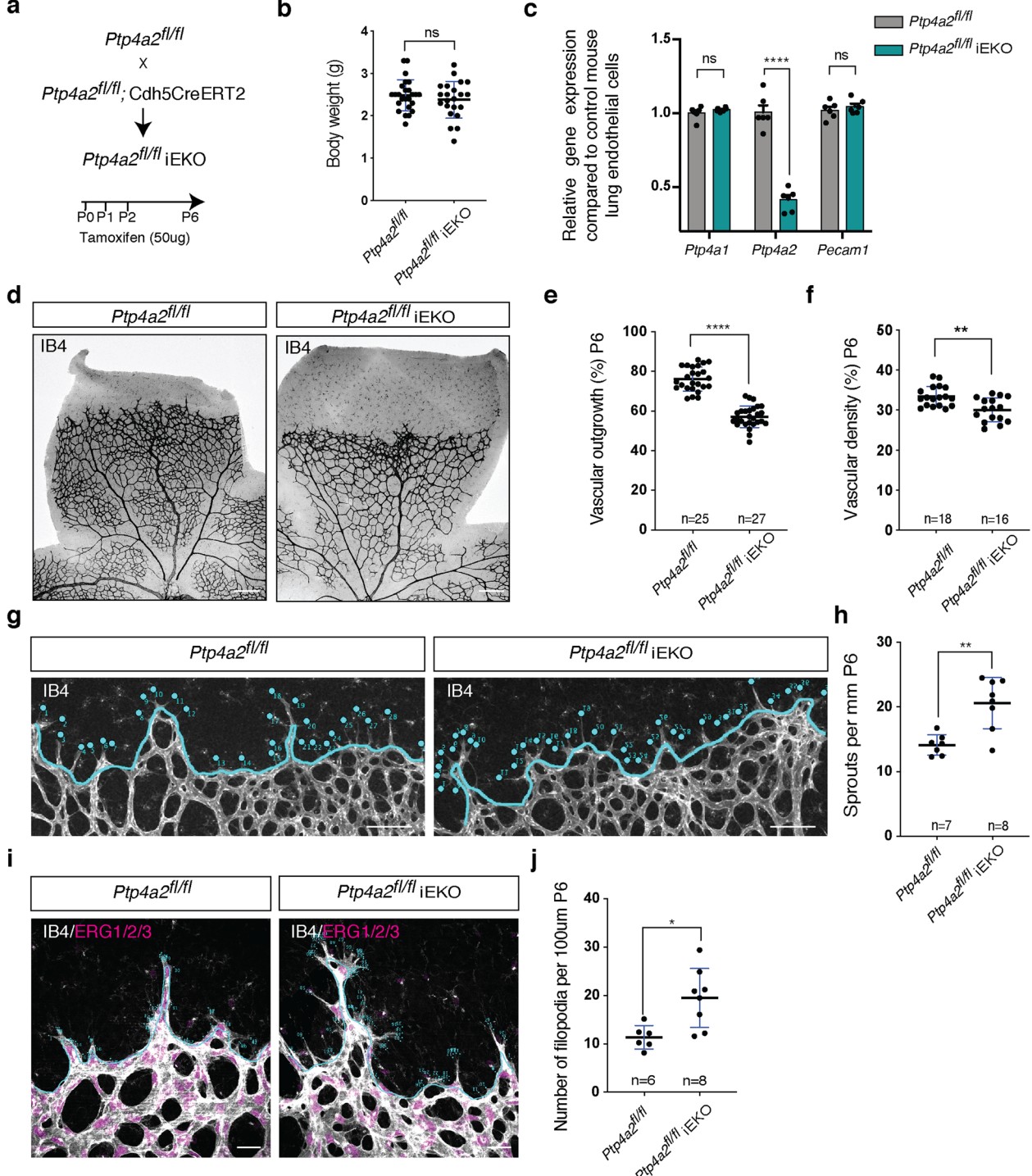

**Fig. 1 Postnatal endothelial *Ptp4a2* deletion impairs vascular development. a** Generation strategy for *Ptp4a2*^fl/fl^iEKO mice and gene deletion through intragastric tamoxifen injection in postnatal mice. **b** Mice weight in P6 *Ptp4a2*^fl/fl^ and *Ptp4a2*^fl/fl^iEKO (Mann–Whitney *U* test: ns nonsignificant *p* value > 0.05). **c** qPCR analysis of mouse lung endothelial cell isolated from P6 *Ptp4a2*^fl/fl^ and *Ptp4a2*^fl/fl^iEKO (two-way ANOVA: ****$p$ value < 0.0001). Each individual data point represents a mouse. **d** Retina whole-mount staining with isolectin B4 (IB4), and quantification of **e** vascular outgrowth and **f** density in P6 *Ptp4a2*^fl/fl^ and *Ptp4a2*^fl/fl^iEKO mice. Each individual data point represents a mouse (average of two retinas) (Mann–Whitney *U* test: **$p$ value < 0.01, ****$p$ value < 0.001). **g** Retina whole-mount staining with isolectin B4 and high magnification of vascular front. **h** quantification of the number of sprouts in P6 *Ptp4a2*^fl/fl^ and *Ptp4a2*^fl/fl^iEKO mice. Each individual data point represents a mouse **i** high magnification of retina whole mounts stained with IB4/ERG1/2/3 and **j** filopodia quantification per sprout (Mann–Whitney *U* test: *$p$ value < 0.05, **$p$ value < 0.01). Each *n* represents individual mouse. Scale bars: **d** 250 µm, **g** 100 µm, **i** 25 µm. Error bars represent mean ± s.e.m.

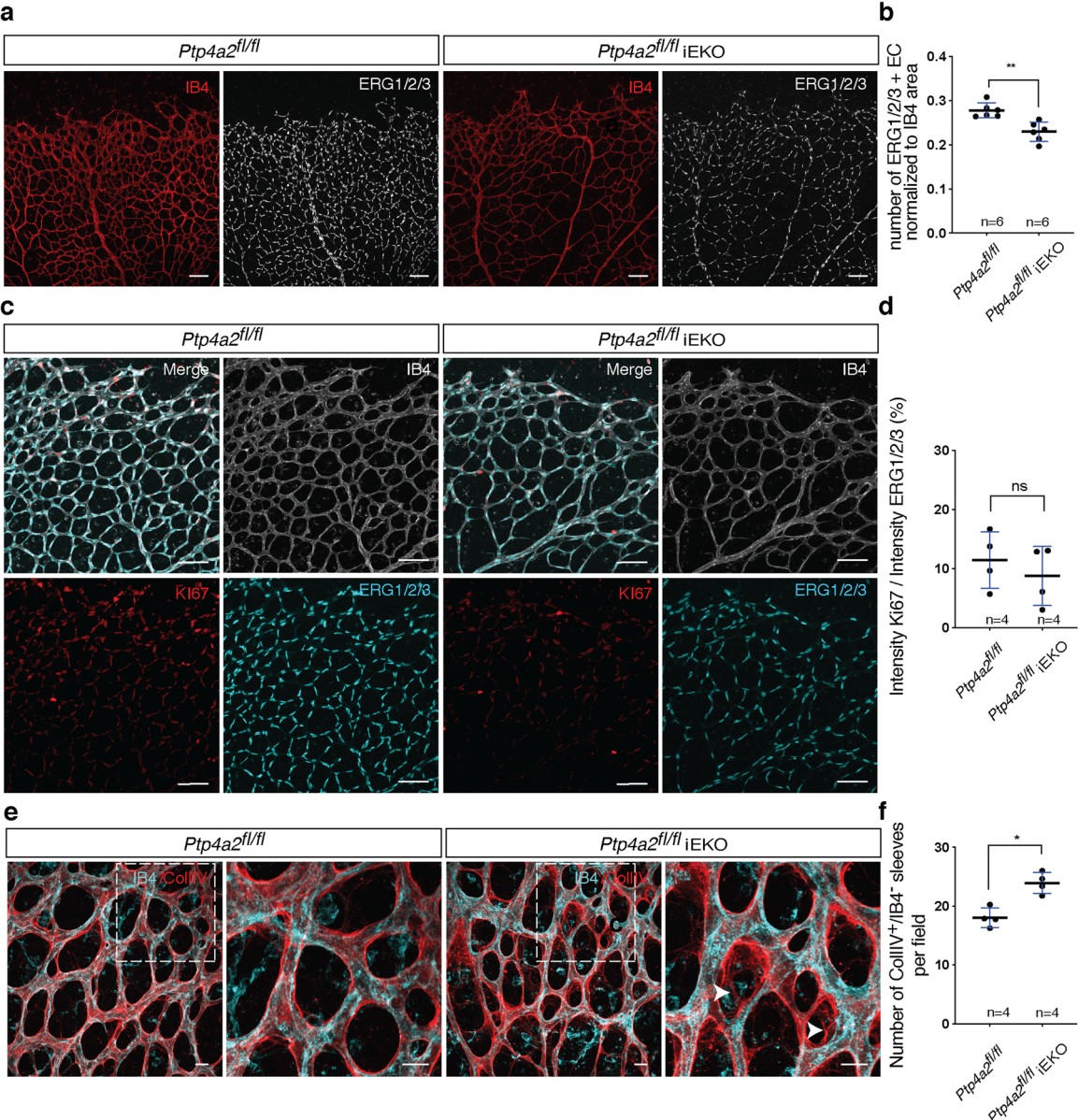

**Fig. 2 *Ptp4a2* postnatal endothelial deletion reduced endothelial cell number and increased vascular regression. a** Retina whole-mount staining with isolectin B4 and ERG1/2/3 and **b** quantification of the number of ERG1/2/3-positive endothelial cells in P6 *Ptp4a2*fl/fl and *Ptp4a2*fl/fliEKO mice. **c** Retina whole-mount staining with isolectin B4 and KI67 and ERG1/2/3, and **d** quantification of proliferative endothelial cells in P6 *Ptp4a2*fl/fl and *Ptp4a2*fl/fliEKO mice. **e** Retina whole-mount staining with isolectin B4 and collagen IV and **f** IB4$^-$/COLIV$^+$ sleeves quantification in P6 *Ptp4a2*fl/fl and *Ptp4a2*fl/fliEKO mice (Mann–Whitney *U* test: *$p$ value < 0.05, **$p$ value < 0.01, ns nonsignificant $p$ value > 0.05). Each individual data point represents a mouse. Scale bars: **a**, **c** 100 μm, **e** 15 μm. Error bars represent mean ± s.e.m.

We next investigated the role of PRL-2 on sprouting angiogenesis, using an in vitro 3D sprouting assay in response to VEGF-A. After 96 h, *PTP4A2* knockdown cells exhibited inadequate and inefficient sprouting angiogenesis compared to control cells (Fig. 5h). Protrusion lengths as well as numbers of cells per protrusion were reduced by 40% and 50%, respectively, in *PTP4A2* knockdown cells compared to control cells (Fig. 5i, j). Interestingly, the total number of these smaller and defective protrusions seen in *PTP4A2* knockdown cells were increased by ~40% compared to control cells (Fig. 5k), hence reflecting the increased number of sprouts seen in vivo in both *Ptp4a2*fl/fliEKO and *Ptp4a2*$^{-/-}$ mice retinas (Figs. 1h and 3h).

**Effect of PRL-2 silencing on VEGF signaling.** Because *PTP4A2* siRNA-treated cells responded differently to VEGF-A

compared to control cells, we next investigated the involvement of PRL-2 in VEGF-A/VEGFR-2 signaling pathways in HUVECs. VEGF-A levels were increased upon *PTP4A2* siRNA treatment (Fig. 6a). We then investigated the VEGFR-2 phosphorylation status after VEGF-A stimulation in control or *PTP4A2* siRNA-treated cells. Surprisingly, upon *PTP4A2* siRNA treatment, a drastic decrease in VEGFR-2-Y1175 phosphorylation was seen after 5 min of VEGF-A stimulation compared to *CTRL* siRNA-treated cells (Fig. 6b, c). However, VEGFR-2-Y951 phosphorylation was not altered (Fig. 6b, d). Consistent with the decreased VEGFR-2-Y1175 phosphorylation, ERK phosphorylation was also reduced by ~50% in *PTP4A2* siRNA-treated cells after 5 min of VEGF-A stimulation (Fig. 6e, f). However, VEGF-A-induced AKT phosphorylation was not affected by PRL-2 downregulation (Fig. 6e, g)

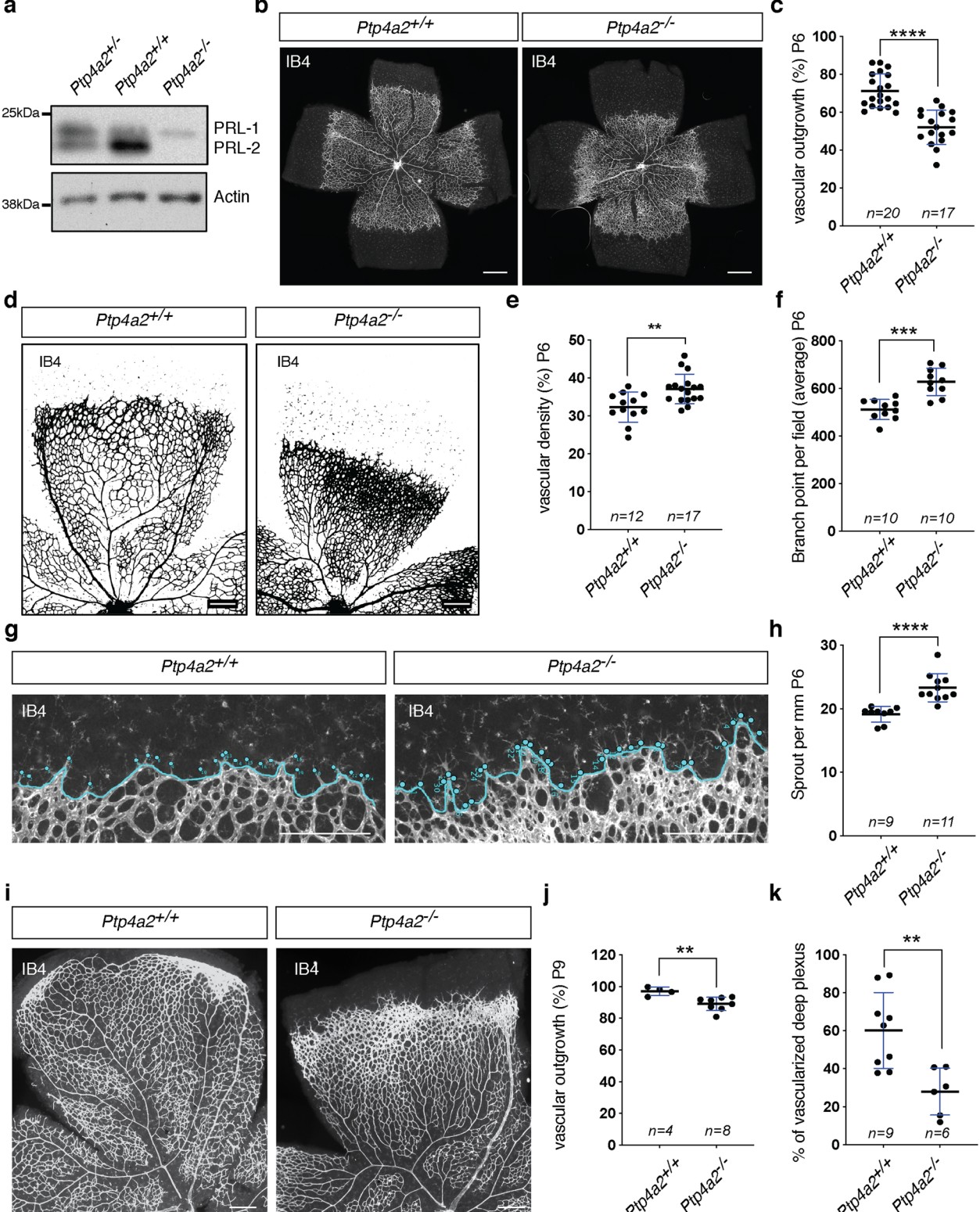

**Fig. 3 Vascular defects in global *Ptp4a2* knockout mice. a** Western blot of protein isolated from the *Ptp4a2*$^{+/-}$ (het), $^{+/+}$ (wt), or $^{-/-}$ (ko) mouse retina. **b** Retina whole-mount staining with isolectin B4 and **c** vascular outgrowth quantification in P6 *Ptp4a2*$^{+/+}$ and *Ptp4a2*$^{-/-}$ mice. Each individual data point represents a mouse (average of two retinas). **d** Retina whole-mount staining with isolectin B4, and quantification of **e** vascular density and **f** branch points in P6 *Ptp4a2*$^{+/+}$ and *Ptp4a2*$^{-/-}$ mice. Each individual data point represents a mouse (average of two retinas). **g** Retina whole-mount staining with isolectin B4 and high magnification of the vascular front and **h** quantification of the number of sprouts in P6 *Ptp4a2*$^{+/+}$ and *Ptp4a2*$^{-/-}$ mice. Each *n* represents an individual mouse. **i** Retina whole-mount staining with isolectin B4 and quantification for **j** vascular outgrowth and **k** deeper plexus vascularization in P9 *Ptp4a2*$^{+/+}$ and *Ptp4a2*$^{-/-}$ mice (Mann–Whitney *U* test: **$p$ value < 0.01, ***$p$ value < 0.001 ****$p$ value < 0.0001). Each individual data point represents a mouse (average of two retinas). Scale bars: **b** 500 μm, **d**, **g**, **i** 250 μm. Error bars represent mean ± s.e.m.

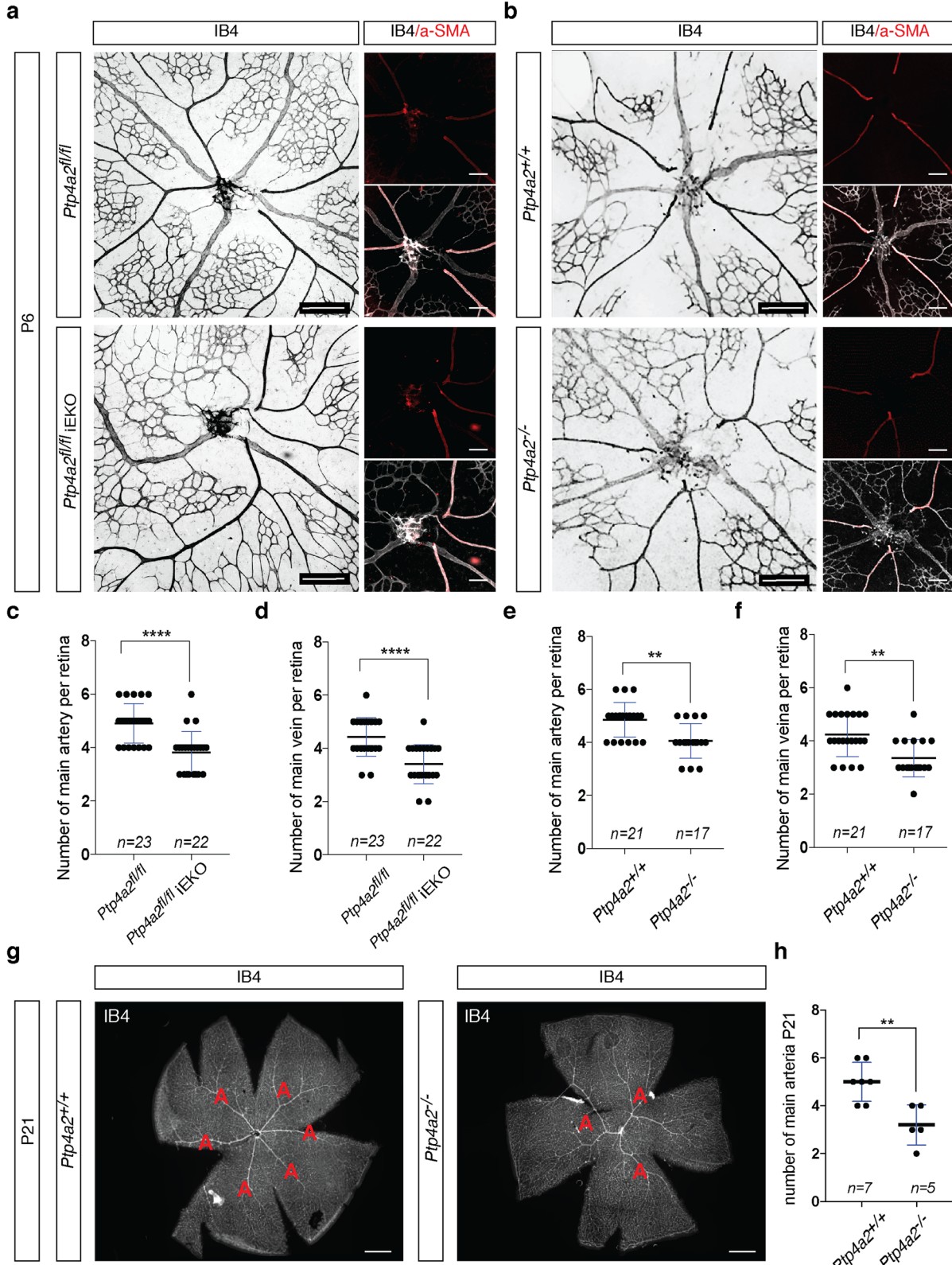

**Fig. 4 PRL-2 is essential for arterial–venous patterning. a** Retina whole-mount staining with isolectin B4 to quantify number of arteries and veins in P6 *Ptp4a2*<sup>fl/fl</sup> and *Ptp4a2*<sup>fl/fl</sup>iEKO mice. **b** Retina whole-mount staining with isolectin B4 and a-SMA to quantify number of arteries and veins in P6 *Ptp4a2*<sup>+/+</sup> and *Ptp4a2*<sup>−/−</sup> mice, and quantification of main vein and artery in **c, d** *Ptp4a2*<sup>fl/fl</sup> and *Ptp4a2* <sup>fl/fl</sup>iEKO mice, or **e, f** *Ptp4a2*<sup>+/+</sup> and *Ptp4a2*<sup>−/−</sup> mice. **g** Retina whole-mount staining with isolectin B4 to see arteries in P21 *Ptp4a2*<sup>+/+</sup> and *Ptp4a2*<sup>−/−</sup> mice, (**h**) and quantification (Mann–Whitney *U* test: **\*p* value < 0.01, \*\*\*\**p* value < 0.0001, ns nonsignificant *p* value > 0.05). Each individual data point represents a mouse. Scale bars: **a**, **b** 250 and 125 µm, **g** 500 µm. Error bars represent mean ± s.e.m.

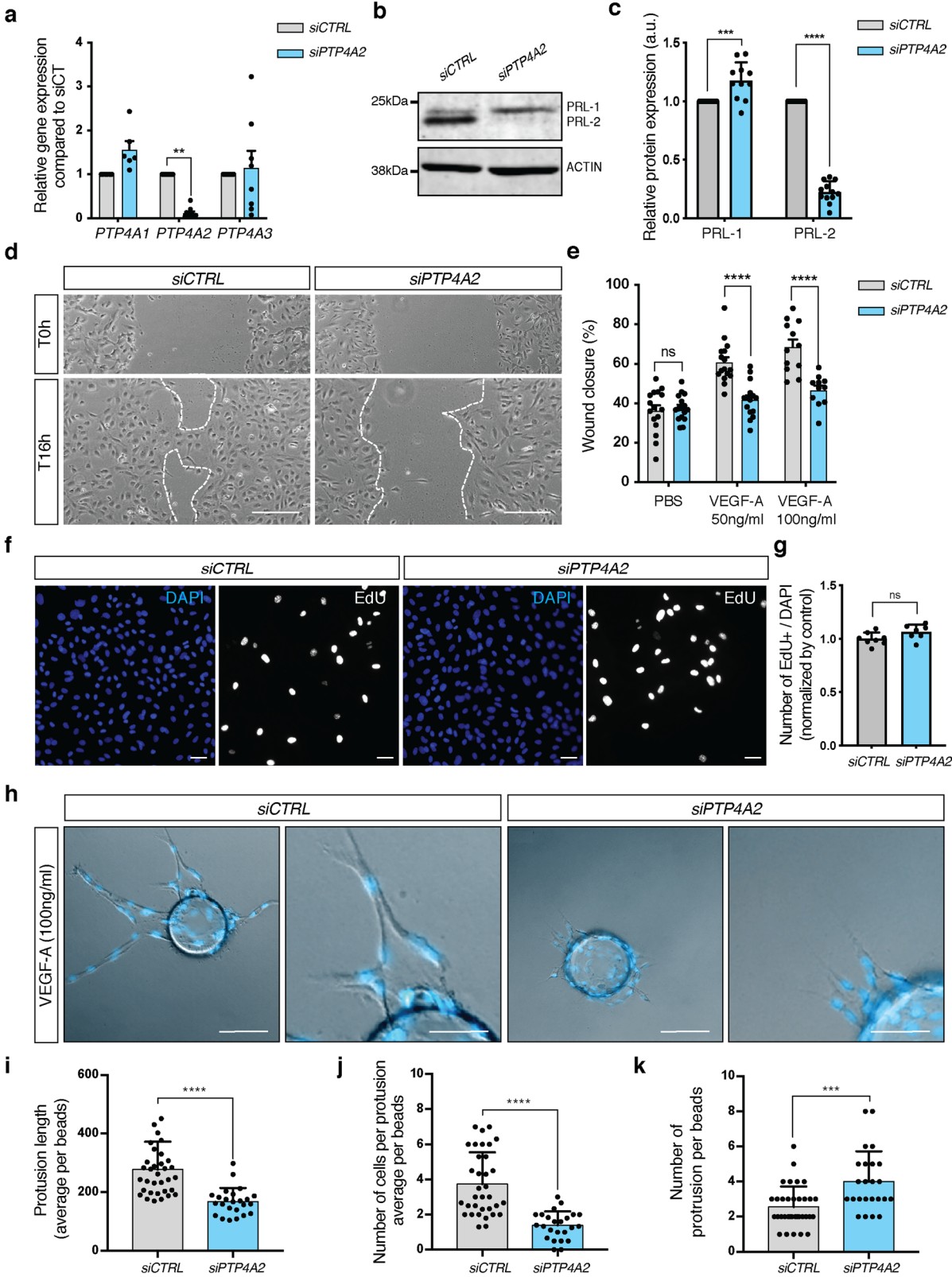

and FGF-2-induced ERK phosphorylation was not affected by PRL-2 knockdown (Supplementary Fig. 6b, c).

**PRL-2 modulates NOTCH-1 signaling in endothelial cells.** Both increased sprouting and defective patterning of arteries and veins (Figs. 1h, 3h and 4), and are hallmarks of impaired

NOTCH-1 signaling[12,60–62]. We demonstrated in HUVEC cells a 50% decrease in the expression of the NOTCH-1 ligand DLL-4, when PRL-2 was downregulated (Fig. 7a). We therefore examined NOTCH-1 cleavage using an antibody which recognizes the intracellular domain (NICD) of NOTCH-1, but only when it is released upon cleavage. Upon VEGF-A stimulation, an almost complete shutdown of NICD release was

**Fig. 5 Effect of PRL-2 silencing on endothelial cell migration and sprouting in vitro. a** qPCR analysis in HUVEC cells after *CTRL* or *PTP4A2* siRNA treatment (at least *n* = 6 independent experiments; two-way ANOVA: *$p$ value < 0.05). **b** Western blot in HUVEC cells, and **c** PRL-1 and PRL-2 protein quantification of western blot shown in **b** after *CTRL* or *PTP4A2* siRNA treatment (*n* = 10 independent experiments; two-way ANOVA: ***$p$ value < 0.001, ****$p$ value < 0.0001). **d** Scratch wound assay performed on HUVEC monolayer in the presence of VEGF-A (50 ng/ml) after *CTRL* or *PTP4A2* siRNA treatment. Images at 0 and 16 h after scratch. **e** Analysis and quantification of the wound closure in HUVEC cells after *CTRL* or *PTP4A2* siRNA treatment in the presence (50 and 100 ng/ml), or absence of VEGF-A (PBS). (Each individual data point represents a biological replicat from *n* = 3 independent experiments, two-way ANOVA: ****$p$ value < 0.0001, ns nonsignificant $p$ value > 0.05). **f**, **g** EdU incorporation assay (8 h) on HUVEC cells after *CTRL* or *PTP4A2* siRNA treatment (Mann–Whitney *U* test: ns nonsignificant $p$ value > 0.05). Each individual data point represents a biological replicat from *n* = 3 independent experiments. **h** HUVEC sprouting assay embedded in 3D fibrinogen gel and stimulated with VEGF-A (100 ng/ml) after *CTRL* or *PTP4A2* siRNA treatment. Quantification of **i** protrusion length, **j** number of cells per protrusion, **k** number of protrusions per beads. Results were expressed as average/ beads. Each individual data point represent a beads; *n* = 3 (Mann–Whitney *U* test: ***$p$ value < 0.001, ****$p$ value < 0.0001). Scale bars: **d** 250 μm, **f** 50 μm, **h** 100 and 50 μm. Error bars represent mean ± s.e.m.

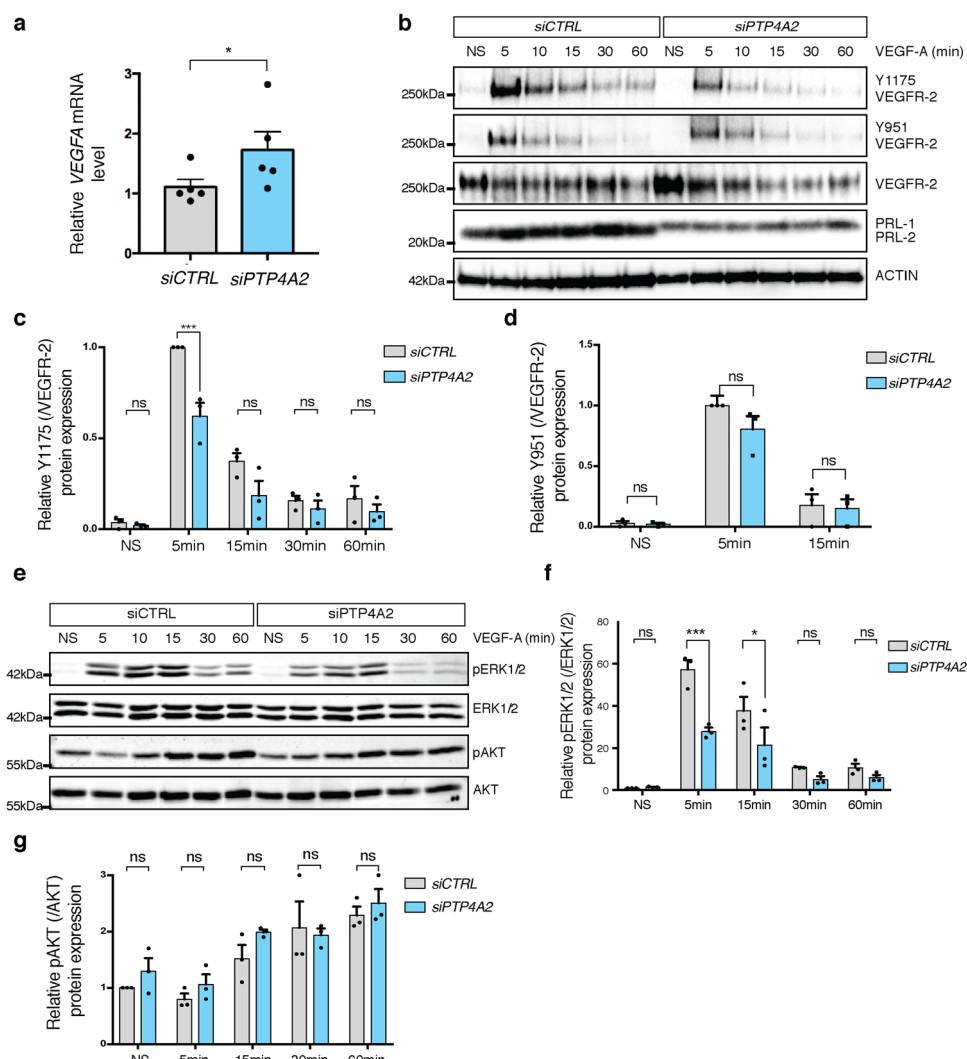

**Fig. 6 Impact of PRL-2 silencing on VEGF signaling in vascular endothelial cells. a** qPCR analysis of *VEGFA* in HUVEC cells after *CTRL* or *PTP4A2* siRNA treatment (Mann–Whitney *U* test: *$p$ value < 0.05; *n* = 5). **b** Effects of VEGF-A treatment on VEGFR-2 phosphorylation (p-Y1175, p-Y951) in siRNA *CTRL* and *PTP4A2* knockdown cells. **c**, **d** Quantification of phosphorylation of VEGFR-2 Y1175 and Y951 normalized to total VEGFR-2 and compared to PBS treated control (*n* = 3 independent experiments; two-way ANOVA: ***$p$ value < 0.001, ns nonsignificant $p$ value > 0.05). **e** Effects of VEGF-A treatment on ERK1/2 and AKT phosphorylation in HUVECs, with *CTRL* siRNA and with siRNA targeting *PTP4A2*. **f**, **g** Quantification of p-ERK normalized to total ERK and pAKT normalized to total AKT (*n* = 3 independent experiments; two-way ANOVA: **$p$ value < 0.01, ***$p$ value < 0.001, ns nonsignificant $p$ value > 0.05). Error bars represent mean ± s.e.m.

observed when *PTP4A2* was knocked down (Fig. 7b, c). Once cleaved, NICD activates transcription factors belonging to the HEY family. We demonstrated that upon DLL-4 stimulation, *HEY2* mRNA expression was decreased in *PTP4A2* siRNA-treated cells (Fig. 7d). It is worth noting that *HES1*, another downstream target of NICD, was also affected by *PTP4A2* silencing (Fig. 7e). Moreover, we confirmed that PRL-2 is an upstream regulator of NOTCH-1 activation by showing that

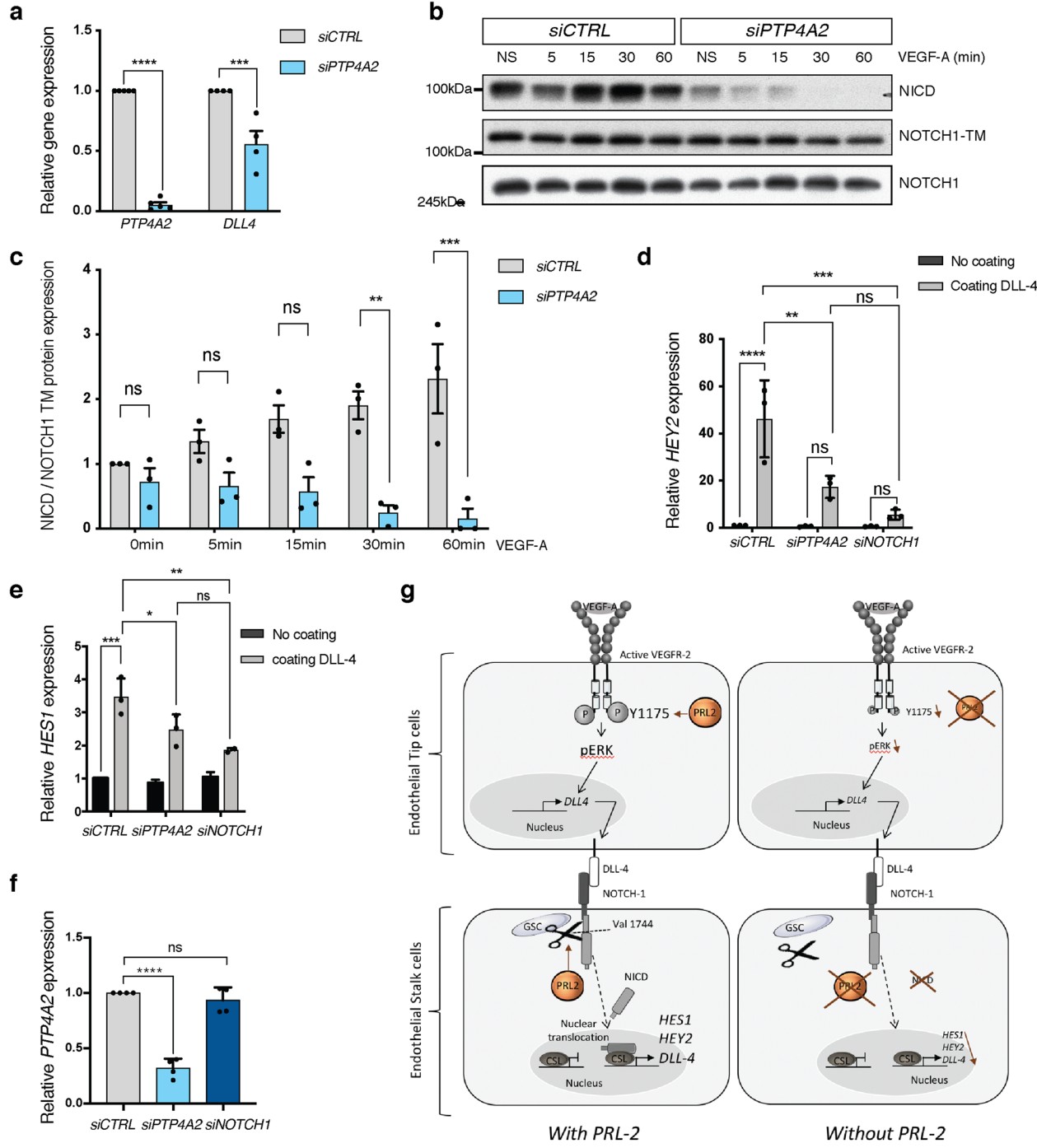

**Fig. 7 PRL-2 regulates NOTCH-1 cleavage. a** qPCR analysis in HUVEC cells after CTRL or PTP4A2 siRNA treatment ($n = 4$ independent experiments; two-way ANOVA: ***$p$ value < 0.001, ****$p$ value < 0.0001). **b** Western blot analysis on HUVEC cells and quantification after *CTRL* or *PTP4A2* siRNA treatment. **c** Quantifications of blots shown in **b** ($n = 3$ independent experiments; two-way ANOVA: **$p$ value < 0.01, ***$p$ value < 0.001, ns nonsignificant $p$ value > 0.05). **d, e** Relative mRNA level of *HEY2* and *HES1*, respectively, assessed by qPCR analysis in HUVEC cells treated with *CTRL*, *PTP4A2*, or *NOTCH1* siRNA and plated overnight on DLL-4-coated plate ($n = 3$ independent experiments; two-way ANOVA: *$p$ value < 0.05 **$p$ value < 0.01, ***$p$ value < 0.001, ****$p$ value < 0.0001, ns nonsignificant $p$ value > 0.05). **f** *Ptp4a2* qPCR analysis in HUVEC cells after *CTRL, PTP4A2*, or *NOTCH1* siRNA treatment ($n = 4$ independent experiments; two-way ANOVA: ****$p$ value < 0.0001, ns nonsignificant $p$ value > 0.05). **g** Schematic representation for PRL-2's role in the vasculature is depicted in the figure. This schematic was created with images adapted from Servier Medical Art licensed under a Creative Commons Attribution 3.0. Error bars represent mean ± s.e.m.

*NOTCH1* siRNA did not have an effect on PRL-2 expression (Fig. 7f). *HEY2* and *HES1* have both been shown to act on arterial–venous specification[63,64], which was indeed altered in both $Ptp4a2^{-/-}$ and $Ptp4a2^{fl/fl}$iEKO mice.

## Discussion

In the present study, we identify PRL-2, a poorly studied member of the PRL family, as an important new regulator of vascular development. Using inducible, endothelial-specific *Ptp4a2*

knockout mice that we generated, we show that the loss of endothelial PRL-2 function leads to impaired postnatal retinal angiogenesis, with effects on angiogenic sprouting, vessel stability, and arterial–venous specification. Global *Ptp4a2* knockouts recapitulated most of these defects, indicating that, in line with its endothelial-enriched expression, PRL-2 mainly functions as a regulator of endothelial cell behavior and vascular morphogenesis. However, defects in angiogenic sprouting were more pronounced in global *Ptp4a2* mutants when compared to mice harboring endothelial-specific deletions, indicating that additional cell types expressing PRL-2 contribute to the vascular development. Our group already demonstrated that PRL-2 is highly expressed in CNS pyramidal neurons, as well as on ependymal cells and photoreceptor cells[65]. Since the retina is part of the CNS, it is possible that additional interference from neurons, microglial cells, or astrocytes present in the developing retina of the *Ptp4a2*$^{-/-}$ mice impacts the vascular phenotype. This is also reinforced by scRNA data that show that *Ptp4a2* is expressed in these cell types (Supplementary Fig. 1).

The mechanism whereby PRL-2 affects endothelial behavior requires further investigation. The PRL family has a weak phosphatase activity due to the presence of an alanine in the active site instead of the conserved serine/threonine residue, which exhibits an essential hydroxyl group[44,45]. Furthermore, their binding to the CNNM-magnesium transporter family via their PTP catalytic site prevent their phosphatase activity[43]. This makes it unlikely that PRLs act via its phosphatase activity. In line with these structural findings, we show that VEGFR-2 signaling and downstream ERK activation is decreased in the absence of PRL function, which is unlikely to be due to its phosphatase activity.

A very important finding of this study is the implication of PRL-2 in NOTCH-1 signaling. A downstream link from NOTCH-1 signaling to PRL-2 was reported in early T cells, whereby NOTCH-1 activation induced the expression of PRL-2 (ref. [66]). However, our finding in the vascular development differs considerably from these findings, since we found an effect of PRL-2 on NOTCH-1 signaling. It has been shown that VEGFR-2 activation in tip cells triggers the stimulation of the NOTCH-1 pathways in stalk cells though DLL-4 upregulation[60,67,68]. When *PTP4A2* is silenced in endothelial cells, we show that, upon VEGF-A stimulation, DLL-4 is downregulated and NOTCH-1 cleavage is impaired. Proteolytic cleavage occurs at the amino acid valine 1744 and is triggered by the γ-secretase complex. It has been demonstrated that Mg$^{2+}$ is an essential cofactor required for γ-secretase activity[69]. We and others have shown that PRL-2 interacts with the CNNM-magnesium sensor and regulates Mg$^{2+}$ flux[59,70,71]. Therefore, PRL-2 may regulate γ-secretase activity indirectly through the regulation of Mg$^{2+}$ flux, although this remains to be fully investigated. Besides, γ-secretase also promotes the cleavage of other proteins, such as Aβ amyloid peptides involved in Alzheimer's disease[69]. Hence, a novel γ-secretase inhibitor, such as PRL-2, could also be of wider clinical interest. The defect in cleavage and formation of NICD through *PTP4A2* downregulation further leads to an inhibition of *HEY2* and *HES1*, two transcription factors implicated in arterial–venous differentiation. Interestingly, *Hey2* knockout mice display vascular growth delays and cardiac malformations associated with lethality around the first 10 days[72–76]. In zebrafish, *HEY2* (grl) knockout further phenocopies our data, since it results in abnormalities in aortic maturation and loss of arterial markers[77]. Thus, the defect in patterning and maintenance of arteries and veins in *Ptp4a2* knockout mice is likely due to a defect in the *HEY2* transcription factor, owing to mis-regulation of the DLL-4/NOTCH-1 pathways. Nevertheless, the dramatic effect on NOTCH-1 cleavage in *PTP4A2* silenced HUVECs does not fully explain the in vivo phenotype, such as observed in the *Dll4*$^{+/-}$ mouse or *Dll4*$^{iΔEC/ΔEC}$ (refs. [12,13,60,61]). Additional compensatory mechanisms may counteract this effect to some extent.

In addition to its effects on NOTCH-1 signaling, we demonstrated that VEGF signaling is also impacted by *PTP4A2* knockdown in endothelial cells. Indeed, *PTP4A2* knockdown led to a decrease in VEGFR-2-Y1175 phosphorylation upon VEGF-A stimulation, and a subsequent decrease of ERK phosphorylation, which can further explain the defect on migration observed in vitro[78] and could explain in part the delay in the vessel formation of the *Ptp4a2* iEKO retina in vivo. Nevertheless, the mechanisms that lead to this effect and its in vivo relevance remains to be addressed. Potential mechanisms are (1) VEGFR-2 receptor desensitization because VEGF-A expression was increased in *PTP4A2* knockdown cells, (2) indirect action of PRL-2 on VEGFR-2 phosphorylation status through the regulation of Mg$^{2+}$ levels, which is critical for receptor kinase activation[44,45], and (3) direct interference of PRL-2 with VEGFR-2 activity.

The link between NOTCH-1, VEGF-A, and PRL-2 signaling remains to be fully elucidated. We have shown that DLL-4 ligand expression is impaired in *PTP4A2* knockdown endothelial cells as well as NOTCH-1 and VEGF-A signaling. Thus, the phenotype observed in *Ptp4a2*$^{fl/fl}$ iEKO mice is most likely due to a combined effect on these different signaling pathways. This is consistent with the known effects of VEGF-A on DLL-4 and NOTCH-1 signaling[60,67,68]. Furthermore, this is important because VEGF-A[79], NOTCH-1 (ref. [80]), DLL-4 (refs. [81,82]), and ERK[83–85] signaling are all involved in arteriogenesis and we indeed observed a defect in the patterning of veins and arteries. A final aspect is a potential role of PRL-2 in vascular stability and pruning since empty basement membranes were increased in *Ptp4a2*$^{fl/fl}$iEKO. It has been shows that NOTCH-1 signaling is involved in vessel pruning and thus PRL-2 may also critically impact on this process via NOTCH-1 signaling[86,87].

*Ptp4a1*, *Ptp4a2*, and *Ptp4a3* are expressed in endothelial cells, and some compensation may occur between them. In particular, endothelial cells express significant amount of *Ptp4a3*, and could also cooperate with *Ptp4a2*/PRL-2 to modulate angiogenesis. The availability of conditional alleles will facilitate further studies to address possible redundancies and cooperation between PRL family members. Furthermore, PRL-3 has been shown to be involved tumor angiogenesis and in placental vascularization[47–49]. It would be interesting to also explore these aspects for PRL-2.

Our results on PRL-2 indicate a completely different function for PRL-2, when compared to the classical phosphatases, such as VE-PTP. PRL-2 has, paradoxically, a vascular promoting role unlike other phosphatases described to date. Figure 7g depicts the general finding of our studies, which highlight the role on VEGF-A and NOTCH-1 signaling. The underlining precise mechanism that account for modulation of endothelial cell signaling is not yet fully elucidated, albeit PRL-2's effect on magnesium metabolism may be involved here. Our findings not only enrich our knowledge in PRL biology, but also provide deeper insights into the regulation vascular development by defining PRL-2 as an important regulator of the developing vasculature.

## Methods

**Genetically engineered mouse models.** All animal procedures were performed according to the Canadian Council on Animal Care ethical regulations and were approved by the McGill University Research and Ethics Animal committee.

*Ptp4a2*$^{-/-}$ mice were described previously[59]. Germline-transmitted *Ptp4a2*$^{+/-}$ mice were backcrossed to C57BL/6 N (Harlan Laboratories) for more than seven generations. Heterozygous breeding pairs were used to obtain littermate wild-type and heterozygous controls. A PRL-2 conditional knockout mouse line was generated using a knockout first conditional-ready targeting construct designed by EUCOMM (EuMMCR repository): (https://www.mousephenotype.org/data/alleles/MGI:1277117/tm88149(L1L2_Pgk_P). The targeting construct was electroporated in 129sv/J ES cell line, and selected targeted clones microinjected in C57Bl6N blastocysts to produce chimeras (GCRC transgenic core facility). Germline

transmission was confirmed, and mice bred with WT C57Bl6 until pure C57Bl6N background, confirmed by MaxBax (Charles River laboratories). In order to remove Neo cassettes flanked by FRT sites, mice were bred with FLPe transgenic mice. For inducible Cre-mediated recombination, *Ptp4a2*$^{fl/fl}$ mice were bred with CDH5-CreERT2 mice[57]. Gene deletion was induced by intragastric injections with 50 μg TAM (Sigma, T5648) to pups at P0, P1, and P2, and mice were sacrificed at P6. The Cre-negative, TAM-treated littermates were use as control mice. Mice of either sex were used in this study.

**Analysis of the retinal vasculature**. Images were analyzed with ImageJ (FIJI). To determine the vascular outgrowth, we calculated the ratio of the retina area and the developing vascular network area. Each individual data point represents a mouse (average of two retinas). To analyze P9 retinas, we quantified the area covered by the deep plexus by determining the area, where the superficial and deep plexus overlap. Each individual data point represents a mouse (average of two retinas). To calculate vessel density, images were converted into 8-bit format and thresholded to obtain a binarized image. In the whole-mount images, the region of interest (ROI) was manually selected, and the area covered by vessel in this ROI was calculated. The vascular area was normalized to the total area of ROI. Each individual data point represents a mouse (average of two retinas). The number of branch point per field was assessed using the Analyze Skeleton tool from the FIJI software. The average of eight fields (250 μm × 250 μm) in capillary area per mice was determined. The sprout number was calculated by manual counting tip cells/filopodia along the sprouting front. These results were normalized to the length of the vascular front in each 10× image. Each individual data point is an average value calculated from three 10× images per mouse. The filopodia number was calculated from four z-max intensity projection of 63× images per mouse, normalized to the length of the vascular front.

For determining the number of ERG1/2/3-positive endothelial cells, the channel was cleaned by using background subtraction and binarized by the user-defined threshold. The number of ERG1/2/3-positive cells was automatically calculated by analyze particles command and normalized to the area of IB4 positive vessel. For each mouse, four 10× images were analyzed and averaged. For proliferation analysis, we assessed KI67 intensity and normalized by the ERG1/2/3 intensity. To do so, the same threshold was applied to all retinas on each channel (KI67, ERG1/2/3). Only cells with a bright signal were included in the quantification since proliferating cells are KI67 high[88]. ROI of ERG1/2/3$^+$ nuclei were selected and applied to both channels. For each mouse four 10× images were analyzed. Neo-vessel retraction was determined by counting the number of COLIV$^+$ and IB4$^-$ sleeves per field. For each mouse, we quantified four fields in the capillary area close to veins and four in the remodeling plexus (field: 350 μm × 350 μm)

**Murine endothelial cell isolation**. Lungs were harvested from 6-day-old mice, shredded and digested with 1 mg/mL collagenase I in Dulbecco's modified Eagle's medium in waterbath at 37 degrees Celsius (°C) for 1 h with shaking every 10 min. The minced tissue was filtered (70 mm pore size, BD Falcon) follow by centrifugation at 1000 × *g* for 10 min at 4 °C. Cells were resuspended in 0.1% BSA, 2 mM EDTA, in DPBS, and were then incubated with G-coated magnetic beads (Invitrogen) pre-coupled with rat anti-mouse PECAM-1 (BD Pharmingen, 553370) for 30 min with agitation. Then, CD31+ cells were isolated using a magnetic rack separator. After three washes with PBS, cells were centrifuged for 5 min at 1000 × *g* and resuspend in RNA lysis buffer. The RNA was purified with the RNeasy Plus Mini Kit (Qiagen, #74134) and reverse transcribed to cDNAs using IScript Reverse Transcriptase III (Bio-Rad, #170–8891) according to manufacturers' instructions.

**Immunohistochemistry**. Retinas were collected at P6, P9, or P21 for whole-mount immunostaining. Eyeball was removed and fixed for 13 min in 4% paraformaldehyde (PFA) and wash three times in PBS 1×. Retinas were dissected and incubated overnight at 4 °C in primary antibodies diluted in blocking buffer (3% BSA, 0.5% Triton X-100, 0.01% Na deoxycholate, 0.02% Na azide in PBS at pH 7.4). After three PBS wash, retinas were incubated 1 h at room temperature with secondary antibodies diluted in PBLEC buffer (1 mM PBS, 1 mM MgCl₂, 1 mM CaCl₂, 0.1 mM MnCl₂, and 1% Triton X-100). Retinas were washed three times and mounted on slide.

Antibodies: COLIV (abcam #ab6586, 0.5 μg/ml), IB4 (Thermo Fisher #I21411, Vector, cat. no. B-1205, 1 μg/ml), KI67 (abcam #ab15580, 1 μg/ml), ERG1/2/3 (abcam #ab196149, 1.5 μg/ml), and α−SMA (sigma #C6198, 1 μg/ml).

**Cell culture and stimulation**. HUAEC were cultured in EGM-2 (Lonza). hCMEC/D3 and HMVEC-D were cultured in EGM-2-MV (Lonza). HUVEC were purchased from Promocell and cultured in endothelial cell growth medium (promocell). HUVEC were used until passages 5 and maintained in collagen-coated plates. For cell stimulation, HUVECs were starved overnight in endothelial cell basal medium (EBM2, promocell) supplemented with 0.1% FBS, and treated with FGF-2 (R&D) or VEGF-A (R&D Systems) during the indicated time and concentration. For DLL-4 stimulation, six-well plates were coated with recombinant human DLL-4 (R&D Systems) diluted into PBS (10 μg/mL), and HUVECs were seeded on it overnight before harvesting.

**siRNA transfection**. The following siRNAs (FlexiTube siRNA) were purchased from Dharmacon: *PTP4A2* (SMART Pool, M-009078-01-0005, GE Healthcare Europe GmbH, Dharmacon), *NOTCH1* (SMART Pool, L-007771-00-0005, GE Healthcare Europe GmbH, Dharmacon), and negative control (SMART Pool, Non-Targeting siRNA#1, GE Healthcare Europe GmbH, Dharmacon). Transfection was performed using lipofectamine RNAimax kit (Thermofisher Scientific, 13778075) in accordance with the manufacturer's instructions, with siRNA at a final concentration of 10 nM in Opti-MEM I (Thermofisher scientific, 31985070). Cells were harvested 48 h after transfection.

**Cell staining and morphology analysis**. Cells were fixed for 15 min at RT in 4% PFA followed by permeabilization with 0.1% Triton X-100 in PBS for 10 min. After washing in PBS, cells were incubated with anti-VE-cadherin AB overnight at 4 °C (Santa Cruz Biotechnology, #SC-6458). Alexa-Fluor-coupled secondary antibodies (Invitrogen), were added to the samples for 1 h at room temperature and cells were counterstained with Hoechst 33342. Samples were mounted in DAKO fluorescent mounting medium. ImageJ software was used to define cell periphery manually, and obtain the total cell area and circularity.

**Fibrin gel angiogenesis assays**. Twenty four hours after siRNA transfection, HUVECs were coated on Cytodex 3 microcarrier beads at a concentration of 400 cells/beads. Twenty four hours after, HUVEC-coated beads were resuspended in 2.5 mg/ml fibrinogen (Sigma-Aldrich) in EBM2 supplemented with 0.15 unit/ml aprotinin (Sigma-Aldrich). Thrombin (Sigma-Aldrich) was added to the fibrinogen/bead solution at the concentration of 10 U/ml. The fibrin gel was clotted for 20 min at 37 °C and complete endothelial cell media supplemented with VEGF (R&D Systems) was added on each well. NIH/3T3 fibroblasts were then platted on top of each well (10,000 cells/well). After 4 days, cells were visualized by doing a DAPI immunostaining.

**Scratch wound healing assay**. Cells were seeded in six-well plate until confluence and then transfected with siRNA for 24 h. Cells were starved overnight with EBM supplemented with 0.5% FBS. A horizontal scratch wound was created across the monolayer using a 200-μl pipette tip. Cells were incubated at 37 °C in EBM2 supplemented with 0.5% FBS and VEGF-A or FGF-2. Pictures were taken just after the scratch (T0 h) and 16 h after (T16 h). Cell migration was calculated using ImageJ software. A minimum of three independent experiment with at least five replicate per condition were analyzed.

**In vitro EdU assay**. We performed the in vitro EdU analysis using Click-iT EdU Cell Proliferation Kit for Imaging (#C10340 Thermo Fisher) on confluent HUVEC monolayers. Forty eight hours after siRNA transfection, the confluent cells were incubated 8 h with EdU and detected according to the staining protocol. For quantification, number of Edu$^+$DAPI$^+$ nucleus were normalized by number of Edu$^-$DAPI$^+$ nucleus on 10× images. Nucleus was quantified by hand in four field (500 × 500 μm) in two to three different biological replicates from n = 3 independent experiment. Each independent experiment was normalized by average control value. On outlayer value was excluded from the analysis.

**RNA isolation, reverse transcription, and quantitative real-time PCR**. Total RNA from cell or retina were extracted using Trizol reagent. A total of 1 μg of total RNA was retrotranscribed into cDNA using the SuperScript III Reverse Transcriptase Kit (Life Technologies). qRT-PCR was performed on a LightCycler 480 with SYBR Green Master Mix according to the manufacturer's instructions. *BACTIN*, and ribosomal protein L13A (*mRpl13A*) were used as housekeeping gene for experiment, using HUVECs or mouse retina lysates, respectively. The list of primer used on this study is displayed on Supplementary data 1.

**Western blots**. Both cell and mouse retina were lysed and homogenized in RIPA buffer (50 mM Tris-HCl pH 7.5, 150 mM NaCl, 1% NP-40, 0.5% sodium deoxycholate, and 0.1% SDS) supplemented with protease and phosphatase inhibitors (Roche Diagnostics). Cell lysates were centrifuged at 16,000 × *g* for 10 min at 4 °C and supernatants were collected. Laemmli buffer was added (62.5 mM Tris pH 6.8, 10% glycerol, 2.5% SDS, and 2.5% β-mercapto-ethanol) and samples were boiled for 5 min at 95 °C. Protein were loaded on SDS–PAGE western blot gel and proteins were electroblotted onto a polyvinylidene difluoride membrane. Membrane blocking was performed using PBS 0.1% Tween supplemented with 5% dry milk for 1 h and membranes were probed overnight at 4 °C, with primary antibodies diluted in blocking buffer (PBS 0.5% BSA).

The following antibodies from cell signaling were used: p44/42 MAPK (p-ERK, cat# 9106, 1:1000), anti-p44/42 MAPK (total ERK, cat# 9102, 1:1000), anti-pAKT (Ser)473 (cat# 4060, 1:1000), anti-AKT (cat# 4691, 1:1000), cleaved NOTCH-1 (Val1744, cat# 4147, 1:500), NOTCH-1 (D6F11, cat# 4380, 1:500), and phospho-specific and total VEGFR-2 (total VEGFR-2, cat# 2479, 1:1000, and Tyr1175, cat# 2478, 1:500, Tyr1214 Tyr951 cat# 2471, 1:500). Rabbit polyclonal anti-β-actin antibodies (cat# A2066, 1:2000), and mouse monoclonal anti-PRL-1/PRL-2 (cat# 05-1583, 1:500) were obtained respectively from Sigma-Aldrich, and EMD

Millipore. Western blot membranes were stripped between phospho- and total protein blotting. Densitometry was quantified using ImageJ.

**Statistics and reproducibility**. Statistical analysis was performed using GraphPad Prism 5 software. The data are presented as mean ± s.e.m. Statistical comparison between two groups was performed by using Mann–Whitney $U$ test. When more than two groups are compared, one or two-way ANOVA test were performed as appropriate, followed by Bonferroni multiple comparison tests. All experiments were repeated at least three times for in vitro experiment. For in vivo experiment, between 4 and 27 mice were analyzed. Sample size ($n$) for each experiment appear in figure legend.

**Reporting summary**. Further information on research design is available in the Nature Research Reporting Summary linked to this article.

## Data availability

The datasets analyzed during the current study in Supplementary Fig. 1 were derived from the following public domain resources: http://tabula-muris.ds.czbiohub.org (Supplementary Fig. 1a), http://betsholtzlab.org/VascularSingleCells/database.html (Supplementary Fig. 1b), and https://markfsabbagh.shinyapps.io/vectrdb/ (Supplementary Fig. 1c). The datasets analyzed in Supplementary Fig. 2a were derived from the sequencing data (accession code GSE86788) from the paper by Jeong et al.[56]. Raw data for graphs presented in the main figures are available in Supplementary Data file 2. The uncropped western blot images are provided in Supplementary Figs. 7 and 8. All other data are available within the manuscript files or from the corresponding author and first author upon reasonable request.

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

## Acknowledgements

This work was supported by a grant from the Association de la Recherche sur le Cancer (ARC) and the Conseil Régional d'Aquitaine to A.B., by Canadian Institutes of Health Research Grants MOP-142497 and FDN-159923 to M.L.T., and by Canadian Institutes of Health Research Grants PJT-165871 to A.D. M.L.T. is a Jeanne and Jean-Louis Lévesque Chair in Cancer Research and holds a Distinguished James McGill Chair from McGill University. M.P. was a recipient of a fellowship from the Ministère de la Recherche (France) and a studentship from the Fondation de la Vie en Rose, Montreal, Canada. We would like to thank Anne Eichmann (Yale Cardiovascular Research Center) for critically reading of the manuscript.

## Author contributions

M.P., A.B., and M.L.T. designed the project from the original idea conceived by A.B., M.L.T., and N.U. M.P., J.S., and A.D. conducted the in vivo experiments. J.S. established the Ptp4a2 iEKO transgenic mouse line. M.P. and K.B. conducted the in vitro experiments. M.P. analyzed the data with the help from K.B., S.H., T.D., A.D., A.B., and M.L.T. The study was supervised by A.B. and M.L.T. The manuscript was written by M.P., K.B., M.L.T., and A.B. and reviewed by all authors.

## Competing interests

The authors declare no competing interests.
