## [Peer Review File · Communications Biology]

Reviewers' comments:

Reviewer #1 (Remarks to the Author):

Manuscript Number: COMMSBIO-20-0877-T

1. Brief summary of the manuscript

The manuscript by Poulet M. et al addresses the role of PRL-2, an atypical protein tyrosine phosphatase, in the regulation of postnatal retinal angiogenesis.

The main findings of the study are:

1. Among the three atypical phosphatases of regenerative liver (PRLs), PRL-2 protein (encoded by the *Ptp4a2* gene) has higher expression in postnatal mouse retina and retinal endothelial cells.
2. Endothelial PRL-2 promotes vessel growth, branching, as well as artery and vein formation in neonatal retina. Partially opposite to *Ptp4a2* inactivation in endothelium, global deletion results in reduced number of arteries and veins and increased vessel density. In vitro, PRL-2 is required for endothelial cell migration.
3. PRL-2 promotes angiogenesis by positively regulating VEGF and DLL4/NOTCH1 signaling pathways.

2. Overall impression of the work

The study sheds light on the previously uncharacterized role of endothelial PRL-2 on sprouting angiogenesis. Overall, the experiments were of good quality, logically addressed and given the results improve our understanding on blood vessel growth, the study should be of interest to many researchers. However, additional insight into the underlying mechanism would strengthen the link between in vitro and in vivo data. Discussion should be improved and aspects that are of peripheral relevance should be taken out.

3. Specific comments, with recommendations for addressing each comment

I am aware that due to the current global situation the authors may not be able to address all the comments.

Major points:

1. The authors describe that among the three PRLs only PRL-2 is more relevant for retinal angiogenesis due to its increased expression throughout vascular development (lines 229-231: "We showed that PRL-2 was the only one PRL family member associated with an increased expression in mouse retina throughout vascular development, indicating the potential implication of this phosphate in developmental angiogenesis."). This needs further clarification.

The homology among PRLs is higher than 70% (Wei M. et al., *Pharmacol Therap*, 2018) and studies showed PRL-3 expression in embryonic blood vessels and tumor endothelium (Li Li et al., *Int J Oncology*, 2016; Guo K et al., *Cancer Res*, 2006; Zimmerman MW et al., *JBC*, 2014). Furthermore, PRL-3 is transcriptionally regulated by VEGF and is important for HUVEC tube formation (Xu J et al., *PLoS One*, 2011). As retinal vasculature develops postnatally and is highly angiogenic, PRL-3 might also play there an important role.

In Fig. 1a legend the authors refer to publication no. 45, but in the corresponding main text, they cite the data from reference no. 53. This should be corrected. The authors need to provide in Methods section the details of data used for Fig. 1a, such as the link to the original data source, the Affymetrix platform used and which probes were analysed for each of the *Ptp4a* genes. As Fig. 1a data are not own data, would rather belong to the Supplementary file.

Analysis of dataset from Jeong HW. et al (*Nature Communic*, 2017), used for Suppl. Fig. 1, indicates that *Ptp4a3* has highest expression among the three *Ptp4a* genes in retinal endothelial cells at P6 (average FPKM: *Ptp4a1*: 15.29; *Ptp4a2*: 28.98; *Ptp4a3*: 35.52). While at P15, *Ptp4a2*

expression is highest (average FPKM: Ptp4a1: 17.74; Ptp4a2: 59.26; Ptp4a3: 29.45), expression of Ptp4a3 was highest again at P21 (average FPKM: Ptp4a1: 11.16; Ptp4a2: 34.91; Ptp4a3: 65.59). Therefore, Suppl. Fig. 1 must be completed with information about Ptp4a3.

Furthermore, in the single-cell transcriptomic analysis done by Vanlandewijck M. et al (Nature, 2018; <http://betsholtzlab.org/VascularSingleCells/database.html>) using FACS sorted Cldn5-GFP endothelial cells from adult mouse brain, Ptp4a3 has rather similar average expression score to Ptp4a2 for different classes of brain endothelial cells presented.

Accordingly, the authors need to rephrase the Discussion paragraph and further assess the expression of PRL-3 in retina and human cell lines. In Fig. 5a the PTP4A3 transcript expression was highly variable between the KD samples. The authors need to comment on that or use another set of primers to answer this question. How is PRL-3 protein level in PTP4A2 KD cells?

PRL-1 protein quantitation is missing from the Fig. 5b graph. In Suppl. Fig. 5b PRL-1 appears increased in siPTP4A2 cells compared to siCT cells. Is PRL-1 increase compensating for the lack of PRL-2?

2. The roles of PRL-2 in endothelial cells and other cellular sources from retina appear to differ in sprouting angiogenesis. While the global deletion of Ptp4a2 increased retinal vessel density, endothelial-specific Ptp4a2 deletion, on the contrary, reduced vascular density. The authors comment in the Discussion part about the possible mechanisms for this difference. The focus of the paper should be rather on the role of endothelial PRL-2 in vascular development and Ptp4a2 global knockout data (Fig. 2) should be moved to the manuscript Supplementary Data.

3. For the characterization of the retinal vascular phenotypes the quantitation details of retinal vasculature parameters presented in Fig. 2, 3, 4 and Suppl. Fig. 2, 3 are missing (e.g. which vascular bed area was chosen for measuring vessel density and number of branching points). The authors should explain in the Methods section how they performed each type of quantitation. What represents individual data point (Fig. 3e, 4b)? Description of n should be added in the figure legend.

4. Furthermore, as Ki67 marks all the cycling cells, to rule out S-phase defects, the authors could perform an EdU/ BrdU pulse-chase in vivo experiment and count the ratio of EdU+/ERG+ per ERG+ ECs from a defined angiogenic front field.

5. The authors found decreased number of ERG+ cells per mutant vessel area (Fig. 4a), which they reasoned to be due to a decreased vessel stability (Fig. 4c). Changes in blood flow, apoptosis and cell migration have been put forward as mechanisms controlling pruning (for a review see Korn C et al., *Dev Cell*, 2015). The authors should discuss the possible role of PRL-2 in vessel stability in relation to the signaling pathways and the mechanisms regulating this process.

6. The authors relate PRL-2 role in endothelial cells to NOTCH and VEGF signaling. Although they propose the major role of PRL-2 is to activate NOTCH signaling (lines 281-283 "Taken together, our article provides compelling evidence for a major regulatory role of PRL-2 in vascular development. PRL-2 seems to act principally on the DLL4/Notch1 pathway but may impact VEGFR2 signaling as well.") the phenotype of Ptp4a2 iECKO retinal vasculature is opposite to NOTCH loss of function phenotypes (Lobov I et al., *PNAS*, 2007; Hellström M et al., *Nature*, 2007; Benedito R et al., *Nature*, 2012).

Given that DLL4 ligand expression is induced by VEGF signaling (Lobov et al., *PNAS*, 2007; Liu ZH et al., *MCB*, 2003; Ubezio B et al., *eLife*, 2016) and that endothelial PRL-2 promotes VEGFR2 activation, the decrease of vascular growth and density observed in Ptp4a2 iECKO mouse retinal phenotype might well be due to a decrease of VEGF and further of NOTCH signaling. The authors should rephrase the text. Retinal immunostaining of ESM1 (a VEGF target) and DLL4 expression could help to confirm their in vitro findings.

The data from Suppl. Fig. 5 should be included in the main figures.

For the WBs quantitation from Suppl Fig. 5 the authors should indicate why the relative protein quantitation was not normalized to the loading control. In Suppl Fig 5e pAkt and Akt bands look similar, albeit with different exposure. It's possible that authors have stripped and reblotted for one of the antibodies. In this situation, the authors should describe in the Methods section.

7. Activated VEGFR2 is phosphorylated on several tyrosine residues, which leads to various endothelial cell responses (Simon M et al., *Nat Rev Mol Cell Biol*, 2016). The authors showed that VEGFR2 phosphorylation at Y1175 but not at Y951 was reduced in PRL-2 KD cells. ERK activation which is downstream of Y1175 phosphorylation was also reduced. Is PLC γ activation affected? The clarification of the signaling upstream of ERK1/2 activation would certainly strengthen the in vitro results.

8. Studies showed that prenylated PRLs are found to the plasma membrane but also in the early endosomes (Zeng Q et al., *JBC*, 2000; Krndija D et al, *JCS*, 2012). Therefore, it is attractive to speculate that PRL-2 may function in VEGFR2 activation, trafficking and downstream signaling activation to control EC behavior. Could this be exploited? The authors could assess whether PTP4A2 KD affects VEGFR2 localization in early endosomes, by co-immunostaining VEGFR2 and EEA1/ Rab5 upon VEGF-A stimulation (e.g. Kofler N et al., *JBC*, 2018).

9. In Fig. 7 the authors investigate the role of PRL-2 on NOTCH signaling. In Fig. 7a n=2 for DLL4 mRNA expression graph. The study requires another experiment. In Fig. 7b the internal loading control is missing from the figure. Please mention for Fig. 7c, d data for how long it was done the DLL4 stimulation of HUVECs?

10. The changes observed in transcript level for DLL4, HEY2/HES1 could be translated at the protein level? The authors should investigate the protein levels of NICD and a downstream target (DLL4, HEY2 or HES1) in basal conditions, upon PTP4A2 KD.

11. The authors observed that Ptp4a2 inactivation results in reduced number of retinal arteries and veins. As also suggested and discussed, PRL-2 could regulate this process through VEGF and NOTCH signaling, which have been shown to regulate arteriovenous specification (for review see Lin FJ et al., *EMBO Rep*, 2007; Fang JF et al., *F1000Res*, 2019). Furthermore, ERK signaling has been shown to regulate not only endothelial cell proliferation and cell migration but also artery formation (Srinivasan R et al., *PLoS One*, 2009; Lanahan A et al., *Dev Cell*, 2010). The authors should better connect PRL-2-VEGF/Notch-pERK-artery formation in their Discussion.

12. Among the multiple roles VEGF has on endothelial cells are regulation of cell migration and barrier formation. In this study, the Ptp4a2 iECKO retinal vascular phenotype could also be attributed to a cell migration defect, due to reduced pERK1/2 or alteration of another signaling downstream of VEGFR2. Please discuss.

In Fig.5C siPTP4A2 KD cells appear more elongated than control cells at 16h post-wounding. Cell size increased? There are cell-cell adhesion defects? To evaluate cell morphology, adhesion and migration defects the authors should immunostain for junctional markers and F-actin. What represents individual data point from Fig. 5c?

13. Furthermore, it is unclear how the quantitation of EdU+ ECs was done in Fig. 5d. This information is missing from the Methods or the figure legend. The authors should quantify the ratio of EdU + cells from total no. of cells in several fields in at least three independent experiments.

Minor points:

1. The title of the manuscript should reflect the newly discovered role of PRL-2.
2. The legends for Fig. 5 and 6 are reversed. The legends for Suppl. Fig. 5 and 6 are missing.
3. The authors should pay attention that protein and gene names to be consistently written

throughout the text, the corresponding figure legend and figure.

4. Molecular weights should be indicated for all Western blotted proteins.

5. In Fig. 2e, it would be more informative to separately display the superficial and the deeper plexus.

6. In Fig. 3c and its legend and text there is a discrepancy regarding the tissue used for the analysis. Please correct.

7. In Fig. 1c legend or in Methods part the authors should explain that the antibody used detects both PRL-1 and PRL-2 proteins. Cat. numbers for antibodies used for immunohistochemistry are missing.

8. For the scratch wound assay the text and the figure indicate 16h as the final time point, whereas in the Methods section was indicated 18h.

Reviewer #2 (Remarks to the Author):

The manuscript by Poulet et al. describe the first functional analysis of a PRL phosphatase, PRL-2/ptp4a2, during vascular morphogenesis. Due to the identification of low enzymatic activity of this family of phosphatases, tissue specific function for this phosphatase in vivo has thus far not been studied in depth. This study shows that PRL-2 is clearly required for angiogenesis by convincing high-quality data. The authors have hereby identified an intriguing novel pathway in vascular biology. I suggest that the manuscript is acceptable for publication if minor comments below are addressed.

- The authors show that there is an increased number of empty collagen sleeves in the EC specific KO. There is no background provided on why this is used in the field as a general read-out of vascular pruning. I suggest the authors add a sentence or two to lead into this result.

- Further mechanistic studies implicate a role for PRL-2 in proliferation via Notch signaling and VEGFR2 signaling. How do these pathways relate to the enhanced vascular pruning?

- Does the antibody used to verify PRL-2 knockdown in vitro (Fig5) also recognize PRL-1? Is this upregulated in the knockdown cells?

- Are there any morphologic changes to knockdown ptp4a2 ECs i.e are there any differences in the expression of VE-cadherin expression or FA markers that can be explained by Notch signaling deficiency and or compromised VEGF signaling?

Reviewer #3 (Remarks to the Author):

The authors investigate the biological function of PRL phosphatases in developmental angiogenesis/morphogenesis. They focus on the specific phosphatase PRL-2, encoded by Ptp4a2, which they suggest is enriched in endothelial cells (ECs) of the developing retina. Global or early postnatally induced EC-specific deletion of Ptp4a2 causes reduced/delayed vascular outgrowth with general patterning defects such as increased regression (reduced stability) and fewer arteries and veins. The authors claim that the phenotype is a consequence of alterations in the Notch pathway and that this lies downstream of PRL-2.

Although the expression data is not novel or even deeply refined this is the first focused study addressing the importance of PRL-2 in the vascular development and to my knowledge the first example of EC-specific deletion of Ptp4a2.

Ptp4a2 is a ubiquitously expressed gene that is slightly enriched in ECs as judged from several recently published data sets acquired through single cell transcriptomics. It is advisable to nuance the EC-selectivity and expression pattern and to direct readers to Sabbagh et al eLife 2018;7:e36187.

The manuscript is clearly written but contain some mismatches in main text vs figures and legends that need to be corrected. As seen below some of the claims are advised to be rephrased.

Importantly the level of experimental details must be improved, especially in terms of image quantification (see comments below).

The authors need to provide detailed descriptions on procedures for quantification of images (vascular density, proliferation, collagen sleeves and deep vascular plexus...). What structures are considered sleeves in contrast to only collagen IV deposits around existing vessels?

Please see further details and comments in the attached review.

PRL-2 is essential for NOTCH1 signaling and retinal angiogenesis

The authors investigate the biological function of PRL phosphatases in developmental angiogenesis/morphogenesis. They focus on the specific phosphatase PRL-2, encoded by *Ptp4a2*, which they suggest is enriched in endothelial cells (ECs) of the developing retina. Global or early postnatally induced EC-specific deletion of *Ptp4a2* causes reduced/delayed vascular outgrowth with general patterning defects such as increased regression (reduced stability) and fewer arteries and veins. The authors claim that the phenotype is a consequence of alterations in the Notch pathway and that this lies downstream of PRL-2.

Although the expression data is not novel or even deeply refined this is the first focused study addressing the importance of PRL-2 in the vascular development and to my knowledge the first example of EC-specific deletion of *Ptp4a2*.

Ptp4a2 is a ubiquitously expressed gene that is slightly enriched in ECs as judged from several recently published data sets acquired through single cell transcriptomics. It is advisable to nuance the EC-selectivity and expression pattern and to direct readers to Sabbagh et al eLife 2018;7:e36187.

The manuscript is clearly written but contain some mismatches in main text vs figures and legends that need to be corrected. As seen below some of the claims are advised to be rephrased.

Importantly the level of experimental details must be improved, especially in terms of image quantification (see comments below).

The authors need to provide detailed descriptions on procedures for quantification of images (vascular density, proliferation, collagen sleeves and deep vascular plexus...). What structures are considered sleeves in contrast to only collagen IV deposits around existing vessels?

Please see further details and comments in the attached review.

Detailed comments

Title

The word "essential" implies that no Notch signalling is possible without PRL-2. Authors show that it affects notch downstream components in vitro but provide no data from mice. This would be required to include such a statement.

Abstract

Row 30, Reads "Thus, our study defines PRL-2 as a major regulator of the vascular development underlying its pro-oncogenic potential reported in several advanced cancers". However, it is a bit of a leap to infer that this would underly the scancer-phenotypes. Perhaps say that it: "may add to its pro-ocogenic...)

Introduction

Line 41: Reads "total absence of vascularisation" which is an over statement, likely more: severely defective vasculature

Line 66: Reads "dual-specificity". It might be useful to explain for what.

Results

Perhaps not call the retina a “model” but rather an anatomical location/organ.
Should read VEGFA not VEGFa, throughout (unless referring to the gene)

Row 83, please clarify that this data derives from published work.

Row 84, clarify that this is whole retina, not enriched ECs.

Row 112: language issues

Rows 114, 115 language

Row 134, Says “strong and significant decrease”, should be sufficient to say “decreased by...” since the percentage is indicated.

Row 196, It is not the NICD “expression” that is investigated, please change.

Figures

Several graphs/plots do not start at 0. Please indicate that the y-axis is distorted to increase clarity (ex Fig 3c,d; 4a...).

1. b. It is unlikely that complete retinal material would represent EC-derived Ptp4a2 as the ECs compose less than 1% of all cells in the retina. This limits the impact of Fig. 1 b. The data would be dramatically improved if instead ECs were isolated and analysed. Alternatively, complete material of iEC Ptp4a2 KO retinas could be analysed to indicate the impact of EC-derived Ptp4a2 on levels from complete retinas, as proof of principle. If this is not possible, I suggest removing this data set and/or find support in recent literature on retinal transcriptomics.

2. Indicate molecular weights in blots

3. Reads “100 ug” which is not aligned with materials and methods, please correct.

d. This picture is repeated in Fig 6a lower panel (mirrored), -replace image.

4. b. pictures and quantification do not seem to match up. Looking in detail there seem to be Ki67 positive signal in all cells that are ERG positive which should not be the case and contradicts quantification that indicate only 10 % proliferative cells in lxlx control condition. Please clarify this and specify how the intensity comparisons were done providing image examples containing original data. How % intensity can be translated into data on proliferation would be helpful.

c. How big is a field?

5. See fig legend, mixup

e. Y-axis legend cannot be correct, likely to be number of protrusions/bead, please comment.

6. See fig legend mixup

7. b. Composes one of the most interesting data with seemingly dramatic results. It is not completely clear how Notch cleavage is affected. To add to this, cells could be stimulated and stained for NICD to show protein distribution. Upon VEGFA stimulation NICD is completely gone suggesting complete blockage of NOTCH signalling in the absence of the phosphatase. Such strong effect on NICD is very intriguing. An experimental repetition would be valuable to clarify solidity.

e. efficiency of Notch1 siRNA-mediated knockdown is not demonstrated, please include.

Lacks “f”. There seem to be only two data points for DLL4 in a, no statistics can hence be applied, please comment.

Figure legends

1. should be “expressed”

a, Should be reference 53.

b. Clarify that it is complete retinal tissue.

c. Indicate size

2. Grammar...

3.

4.

5 (should be 6). Swapped places with Fig. 6. Arteries and veins do form but they are fewer. More accurate to state: PRL-2 I essential for patterning of veins and arteries.

6. (should be 5)

7. Title includes data that is not shown in the figure, please correct. No panel for “f” as referred to.

-REBUTAL LETTER

We would like to thank all the reviewers for their helpful comments and suggestions, which enabled us to improve the manuscript substantially.

Below is a point-by-point reply to each reviewer's comments.

Reviewer #1 (Remarks to the Author)

Information to the reviewer: A detailed description of the modification of the Figures (figure modification sum-up) is added at the end of the reply to the reviewers.

COMMENT 1. Brief summary of the manuscript

The manuscript by **Poulet M.** et al addresses the role of PRL-2, an atypical protein tyrosine phosphatase, in the regulation of postnatal retinal angiogenesis.

The main findings of the study are:

1. Among the three atypical phosphatases of regenerative liver (PRLs), PRL-2 protein (encoded by the *Ptp4a2* gene) has higher expression in postnatal mouse retina and retinal endothelial cells.
2. Endothelial PRL-2 promotes vessel growth, branching, as well as artery and vein formation in neonatal retina. Partially opposite to *Ptp4a2* inactivation in endothelium, global deletion results in reduced number of arteries and veins and increased vessel density. *In vitro*, PRL-2 is required for endothelial cell migration.
3. PRL-2 promotes angiogenesis by positively regulating VEGF and DLL4/NOTCH1 signaling pathways.

COMMENT 2. Overall impression of the work

The study sheds light on the previously uncharacterized role of endothelial PRL-2 on sprouting angiogenesis. Overall, the experiments were of good quality, logically addressed and given the results improve our understanding on blood vessel growth, the study should be of interest to many researchers. However, additional insight into the underlying mechanism would strengthen the link between *in vitro* and *in vivo* data. Discussion should be improved and aspects that are of peripheral relevance should be taken out.

Reply: We appreciate the positive assessment of our manuscript by the reviewer. New experiments were added and discussed in the point-by-point reply. The manuscript was reorganized and the discussion improved according to the reviewer's comments.

COMMENT 3. Specific comments, with recommendations for addressing each comment
I am aware that due to the current global situation the authors may not be able to address all the comments.

Reply: We thank the reviewer for understanding the difficulties in performing research in particular animal experiments, during the COVID-19 crisis. We were able to respond to most of the reviewer's comments, however, some suggested experiments could not be performed. The reviewer also mentioned in his comments that this might be the case.

Major points:

COMMENT 4. The authors describe that among the three PRLs only PRL-2 is more relevant for retinal angiogenesis due to its increased expression throughout vascular development (lines 229-231:“ We showed that PRL-2 was the only one PRL family member associated with an increased expression in mouse retina throughout vascular development, indicating the potential implication of this phosphate in developmental angiogenesis.”). This needs further clarification.

The homology among PRLs is higher than 70% (Wei M. et al., *Pharmacol Therap*, 2018) and studies showed PRL-3 expression in embryonic blood vessels and tumor endothelium (Li Li et al., *Int J Oncology*, 2016; Guo K et al., *Cancer Res*, 2006; Zimmerman MW et al., *JBC*, 2014). Furthermore, PRL-3 is transcriptionally regulated by VEGF and is important for HUVEC tube formation (Xu J et al., *PLoS One*, 2011). As retinal vasculature develops postnatally and is highly angiogenic, PRL-3 might also play there an important role.

Reply:

We have now taken into account the reviewers suggestions, cited further relevant articles with regard to vascular expression of all PRLs and performed additional analyses. We also discussed the potential role of other PRLs in the line 318-323 of the discussion. When the 3 PRLs are analyzed more in detail, the following conclusion can be drawn: All PRLs share at least 75% amino acid sequence identity¹. However, PRL-1 and PRL-2 are the most similar in terms of their structure. The mechanism of action of PRL-3 described in the literature seems to be very different from PRL-1 or PRL-2. PRL-3 has been shown to act on phosphoinositides² which has not been demonstrated for PRL-1 or PRL-2. In our article we focused primarily on the role of PRL-2 developmental angiogenesis in an established model, the retina) using conditional endothelial *Ptp4a2*^{-/-} mice and provided *in vitro* data and signaling studies. We agree with the reviewer that there is also a significant amount of *Ptp4a3* expressed in endothelial cells and both PRLs could cooperate to modulate angiogenesis. This applies also to the retinal vasculature since it develops postnatally and is highly angiogenic. Furthermore, as the reviewer indicated, a role in tumor angiogenesis has been shown for PRL3 and in placental vascularization³⁻⁵.The availability of conditional alleles will facilitate further studies to address possible redundancies and cooperation between PRL family members.

COMMENT 5: In Fig. 1a legend the authors refer to publication no. 45, but in the corresponding main text, they cite the data from reference no. 53. This should be corrected. The authors need to provide in Methods section the details of data used for Fig. 1a, such as the link to the original data source, the Affymetrix platform used and which probes were analyzed for each of the *Ptp4a* genes. As Fig. 1a data are not own data, would rather belong to the Supplementary file.

Reply: We are grateful for the thorough evaluation of the manuscript by the reviewer and we apologize for this reference error. Figure 1A has been replaced by more comprehensive analyses using different data sets (Tabula Muris: <https://tabula-muris.ds.czbiohub.org/>⁶ ; searchable database: <http://betsholtzlab.org/VascularSingleCells/database.htm>⁷ ; Trans-omics Resource

Database (VECTRDB)(<https://markfsabbagh.shinyapps.io/vectrdb/>⁸). Thus, the corresponding reference has been deleted.

COMMENT 6: Analysis of dataset from Jeong HW. et al (Nature Communication, 2017), used for Suppl. Fig. 1, indicates that *Ptp4a3* has highest expression among the three *Ptp4a* genes in retinal endothelial cells at P6 (average FPKM: *Ptp4a1*: 15.29; *Ptp4a2*: 28.98; *Ptp4a3*: 35.52). While at P15, *Ptp4a2* expression is highest (average FPKM: *Ptp4a1*: 17.74; *Ptp4a2*: 59.26; *Ptp4a3*: 29.45), expression of *Ptp4a3* was highest again at P21 (average FPKM: *Ptp4a1*: 11.16; *Ptp4a2*: 34.91; *Ptp4a3*: 65.59). Therefore, Suppl. Fig. 1 must be completed with information about *Ptp4a3*.

Furthermore, in the single-cell transcriptomic analysis done by Vanlandewijck M. et al (Nature, 2018; <http://betsholtzlab.org/VascularSingleCells/database.html>) using FACS sorted Cldn5-GFP endothelial cells from adult mouse brain, *Ptp4a3* has rather similar average expression score to *Ptp4a2* for different classes of brain endothelial cells presented.

Reply: we have taken into account the reviewer's comments and performed some additional analyses. We added, according to the reviewer's suggestions, the data on the *Ptp4a3* (Bulk RNA sequencing of retinal endothelial cells, to the figure (now supplementary Fig 2a)⁹. A strong increase in *Ptp4a2* was seen at early time points (P6-P15) albeit a decrease was observed at p21. This is different to *Ptp4a1* and *Ptp4a3*. For *Ptp4a1* low expression levels were detected during retina development, whereas *Ptp4a3* expression only peaks at late time-points (p21) (Figure 1).

Figure 1: Analysis of *Ptp4a1-3* in the online bulk RNA sequencing data set from P6 to P21 mouse retinal endothelial cell⁹.

Second, as suggested by the reviewer, we analyzed the single cell sequencing data set by Betsholtz and collaborators⁷ on brain endothelial cells, because the retina is also part of the central nervous system. The data from adult mouse brain indicate that *Ptp4a2* is

the most expressed in some endothelial cell clusters (EC1) in the nervous system. However, *Ptp4a3* seems to be more expressed in mural cells (vascular smooth muscle and pericytes) (Figure 2).

Figure 2: Analysis of *Ptp4a1-3* expression in the single cell RNA sequencing dataset. (searchable database: <http://betsholtzlab.org/VascularSingleCells/database.html>)⁷. Average expression in each cluster [Brain data] PC - Pericytes; SMC - Smooth muscle cells; MG - Microglia; FB - Vascular fibroblast-like cells; OL - Oligodendrocytes; EC - Endothelial cells; AC - Astrocytes; v - venous; capil - capillary; a - arterial; aa - arteriolar; 1,2,3- subtypes.

Finally, we analyzed an additional single cell data set of the various cell types in the brain. (Tabula Muris: <https://tabula-muris.ds.czbiohub.org/>)⁶. As it can be seen from the violin plots below, *Ptp4a1* is poorly expressed (mean of 0.58), *Ptp4a3* has an intermediate expression level (mean of 2.94) and *Ptp4a2* has the highest expression (mean of 4.13). In addition, *Ptp4a2* is more expressed in endothelial cells than in pericytes or in any other cell type. This is the opposite for *Ptp4a3*.

Figure 3: Analysis of *Ptp4a1-3* expression single-cell transcriptomic data from the model organism *Mus musculus* (searchable database *Tabula Muris* <https://tabula-muris.ds.czbiohub.org/>)⁶. *Ptp4a1-3* expression for each cell type sorted by FACs in brain (non-myeloid cells) can be visualized with violin plots.

We have also included the analysis of an additional scRNA data set from the Vascular Endothelial Cell Trans-omics Resource Database (VECTRDB) (<https://markfsabbagh.shinyapps.io/vectrdB/>)⁸. Different endothelial cell types were analyzed in the brain by scRNAseq. This latter analysis provides information on *Ptp4a2* expression in the different endothelial cell types (included in the manuscript as supplementary Figure 1c and Figure 4 in the letter). *Ptp4a2* is globally more uniformly expressed in all endothelial cell types.

Figure 4: Analysis of *Ptp4a1-3* expression in Vascular Endothelial Cell Trans-omics Resource Database (VECTRDB) (<https://markfsabbagh.shinyapps.io/vectrdb/>). *Ptp4a1-3* gene expression in the developing CNS at single cell resolution. Single-cell RNA-seq on 3,946 FACS-purified GFP-positive endothelial cells from a P7 Tie2-GFP mouse brain. This dataset allows assessment of *Ptp4A1-3* expression in arterial, venous, capillary, mitotic, and tip subtypes of brain endothelial cells.

COMMENT 7: Accordingly, the authors need to rephrase the Discussion paragraph and further assess the expression of PRL-3 in retina and human cell lines. In Fig. 5a the *Ptp4a3* transcript expression was highly variable between the KD samples. The authors need to comment on that or use another set of primers to answer this question. How is PRL-3 protein level in *Ptp4a2* KD cells?

Reply: We have conducted at least 6 independent siRNA experiments to analyze *Ptp4a1-3* expression by qPCR. It is true that expression level of *Ptp4a3* is variable which may be due to a different growth state of the cells. However, the mean expression is not significantly increased. The PRL-3 protein level was not determined because, to our knowledge, anti-PRL-3 antibodies perform poorly in western blots.

COMMENT 8: PRL-1 protein quantitation is missing from the Fig. 5b graph. In Suppl. Fig. 5b PRL-1 appears increased in si*Ptp4a2* cells compared to siCtrl cells. Is PRL-1 increase compensating for the lack of PRL-2?

Reply: As suggested by the reviewer, we have performed quantification of PRL-1 protein after *Ptp4a2* knock-down in HUVECs. PRL-2 protein expression was strongly decreased by 80 % and a slight increase (20 %) in PRL-1 was seen in western blots after *Ptp4a2* knock-down when a large number of blots were analyzed (n=10). This has now been added to the manuscript (Figure 5 of the rebuttal letter, figure 4c of the manuscript). PRL-1 may compensate the activity of PRL-2 to some extent, however *Ptp4a2* knock-down cells still exhibit strong phenotypic modifications and changes in cell signaling.

Figure 5: Quantification of PRL-1 and PRL-2 expression assessed by Western blot analysis (n=10 experiments)

COMMENT 9: The roles of PRL-2 in endothelial cells and other cellular sources from retina appear to differ in sprouting angiogenesis. While the global deletion of *Ptp4a2* increased retinal vessel density, endothelial-specific *Ptp4a2* deletion, on the contrary, reduced vascular density. The authors comment in the Discussion part about the possible mechanisms for this difference. The focus of the paper should be rather on the role of endothelial PRL-2 in vascular development and *Ptp4a2* global knockout data (Fig. 2) should be moved to the manuscript Supplementary Data.

Reply: We agreed with the reviewer suggestion and have modified the article by placing the *Ptp4a2* iEKO mouse retina data in figure 1 (New Figure 1). We prefer to keep the full body *Ptp4a2*^{-/-} as a main figure and not placing it in supplemental data, since most of the phenotype observe for *Ptp4a2* iEKO was also conserved in the full body ko. The comparison between *Ptp4a2* iEKO and *Ptp4a2*^{-/-} also suggested the participation of non-endothelial PRL2 function to vascular development.

COMMENT 10: For the characterization of the retinal vascular phenotypes the quantitation details of retinal vasculature parameters presented in Fig. 2, 3, 4 and Suppl. Fig. 2, 3 are missing (e.g. which vascular bed area was chosen for measuring vessel density and number of branching points). The authors should explain in the Methods section how they performed each type of quantitation. What represents

individual data point (Fig. 3e, 4b)? Description of n should be added in the figure legend.

Reply: The methods section has been revised and improved by including numerous information and quantifications details as requested by the reviewer. In the first version of the manuscript, individual data point in Figure 3e and 4b represented each microscope field analyzed. In the revised manuscript (now Figure 1f and 2d), individual data point in these figures represent now the average of the different fields from an individual mouse (each data point=one mice). The significance of each individual data point (n) was added in figure legend.

COMMENT 11: Furthermore, as Ki67 marks all the cycling cells, to rule out S-phase defects, the authors could perform an EdU/ BrdU pulse-chase *in vivo* experiment and count the ratio of EdU+/ERG+ per ERG+ ECs from a defined angiogenic front field.

Reply: Due to the COVID-19 crisis, animal experimentation has slowed down. In addition, we had reduced our *Ptp4a2* iEKO mouse colony to the minimum and it will take several months to expand them and to do the required staining. Thus, we would prefer not to wait for this experiment to be included in the revision of this manuscript. The reviewer also stated that in the current situation it is understandable that not all revision can be done.

COMMENT 12: The authors found a decreased number of ERG+ cells per mutant vessel area (Fig. 4a), which they reasoned to be due to a decreased vessel stability (Fig. 4c). Changes in blood flow, apoptosis and cell migration have been put forward as mechanisms controlling pruning (for a review see Korn C et al., Dev Cell, 2015). The authors should discuss the possible role of PRL-2 in vessel stability in relation to the signaling pathways and the mechanisms regulating this process.

Reply: A potential role of PRL-2 in vascular stability and pruning may be due to altered VEGF and/or Notch signaling regulated by PRL-2. It has been shown that withdrawal of VEGF-A leads to vessel regression. PRL-2 renders vessels less responsive to VEGF and thus may favor vessel regression. NOTCH-1 was shown to be involved in vessel pruning^{10,11}. Stabilization of post-natal vessels depend on the crosstalk between the WNT/ β -catenin pathway with the Delta-like 4 (DLL4)/Notch pathway¹⁰. Thus, PRL-2 may control stability and pruning also via this mechanism.

It is known that γ -secretase activity is highly sensitive magnesium¹². Since PRL-2 is involved in magnesium metabolism¹³, its effect on NOTCH may be indirect through the regulation of γ -secretase activity. This has not yet been clearly addressed but should be part of a follow-up study. Furthermore, we have investigated apoptosis *in vitro* via cleaved caspase staining and no differences were detected in *Ptp4a2* knock-down cells when compared to control cells (Figure 6).

Figure 6: quantification of the number of cleaved caspase positive cells normalized by DAPI after siCtrl or siPtp4a2 treatment

COMMENT 13: The authors relate PRL-2 role in endothelial cells to NOTCH and VEGF signaling. Although they propose the major role of PRL-2 is to activate NOTCH signaling (lines 281-283 “Taken together, our article provides compelling evidence for a major regulatory role of PRL-2 in vascular development. PRL-2 seems to act principally on the DLL4/Notch1 pathway but may impact VEGFR2 signaling as well.”) the phenotype of *Ptp4a2* iECKO retinal vasculature is opposite to NOTCH loss of function phenotypes (Lobov I et al., PNAS, 2007; Hellström M et al., Nature, 2007; Benedito R et al., Nature, 2012). Given that DLL4 ligand expression is induced by VEGF signaling (Lobov et al., PNAS, 2007; Liu ZH et al., MCB, 2003; Ubezio B et al., eLife, 2016) and that endothelial PRL-2 promotes *VEGFR-2* activation, the decrease of vascular growth and density observed in *Ptp4a2* iECKO mouse retinal phenotype might well be due to a decrease of VEGF and further of NOTCH signaling. The authors should rephrase the text. Retinal immunostaining of ESM1 (a VEGF target) and DLL4 expression could help to confirm their *in vitro* findings.

Reply: We are grateful for the reviewer’s comment which raised a very interesting point. We agree that, the dramatic effect on Notch signaling *in vitro* cannot fully account alone for the phenotype observed in the retinas of *Ptp4a2* iEKO mice such as observed in the $DLL4^{+/-}$ or $DII4^{i\Delta EC/\Delta EC}$ mice^{14,15}. The mechanism of action explaining the phenotype seen in the *Ptp4a2* iEKO mouse retina is still not completely elucidated and involves both NOTCH and VEGF signaling. There is surely a common theme underlining PRL-2 mechanism of action which may target simultaneously NOTCH and VEGF signaling. One may speculate that Mg^{2+} metabolism is involved. This has been now discussed and rephrased in the manuscript. (line 294-297)

COMMENT 14: The data from Suppl. Fig. 5 should be included in the main figures. For the WBs quantitation from Suppl Fig. 5 the authors should indicate why the relative protein quantitation was not normalized to the loading control.

In Suppl Fig 5e pAkt and Akt bands look similar, albeit with different exposure. It's possible that authors have stripped and reblotted for one of the antibodies. In this situation, the authors should describe in the Methods section.

Reply: As the reviewer suggested, we included the supplementary figure 5 as principal figure 6 in the revised version of the manuscript. For all western-blot quantification, we decided to normalize to the more relevant loading control. All phospho-protein (pERK, pAKT or pVEGFR-2) were normalized to the total level of the same protein (ERK, AKT, VEGFR-2, accordingly). This is the classical method for phospho-protein quantification. Indeed, since these proteins are highly subjected to internalization and degradation upon stimulation by various growth factor, any phosphorylated protein needs to be normalized to its total protein. Western blot membranes were stripped between phospho- and total protein blotting. This is now added in the methods section as the reviewer suggested. (line 481)

COMMENT 15: Activated VEGFR2 is phosphorylated on several tyrosine residues, which leads to various endothelial cell responses (Simon M et al., Nat Rev Mol Cell Biol, 2016). The authors showed that VEGFR2 phosphorylation at Y1175 but not at Y951 was reduced in PRL-2 KD cells. ERK activation which is downstream of Y1175 phosphorylation was also reduced. Is PLC γ activation affected? The clarification of the signaling upstream of ERK1/2 activation would certainly strengthen the *in vitro* results.

Reply: As the reviewer suggested, we have performed analysis of PLC- γ expression after *Ptp4a2* knock-down. This is shown in the figure below. It can be seen that PLC- γ phosphorylation is reduced.

Figure 7: Western blot of phospho-PLC γ after *Ptp4a2* siRNA treatment in HUVEC cells

COMMENT 16: Studies showed that prenylated PRLs are found to the plasma membrane but also in the early endosomes (Zeng Q et al., JBC, 2000; Krndija D et al, JCS, 2012). Therefore, it is attractive to speculate that PRL-2 may function in *VEGFR-2* activation, trafficking and downstream signaling activation to control EC behavior. Could this be exploited? The authors could assess whether *Ptp4a2* KD affects *VEGFR-2* localization in early endosomes, by co-immunostaining *VEGFR-2* and EEA1/ Rab5 upon VEGF-A stimulation (e.g. Kofler N et al., JBC, 2018).

Reply: We agree with the reviewer that trafficking studies could be performed in this context. However, this will not add significant information on the precise mechanisms of action of PRL-2 in endothelial cells. We believe such experiments are out of the scope of the current manuscript and require an extensive follow-up investigation such as trafficking studies of a potential PRL-2/substrate complex. We hope that the reviewer

agrees. We have nevertheless performed an experiment to see whether VEGFR2 could directly or indirectly physically interact with PRL-2. As seen in the IP / Western blot below, we could not detect an interaction between both proteins.

Figure 8: IP / Western experiment to visualize a potential complex between PRL-2 and VEGFR2. ECs were stimulated with VEGF-A and cell extracts were immunoprecipitated using an anti-VEGFR2 antibody. After transfer onto nitrocellulose membrane, the membrane was probed using the anti-PRL-2 antibody.

COMMENT 17: In Fig. 7 the authors investigate the role of PRL-2 on NOTCH signaling. In Fig. 7a n=2 for *DLL4* mRNA expression graph. The study requires another experiment. In Fig. 7b the internal loading control is missing from the figure. Please mention for Fig. 7c, d data for how long it was done the *DLL4* stimulation of HUVECs?

Reply: We agreed with the reviewer's comment and apologize for the mistake. The graph depicting *Dll4* mRNA expression (Fig 7A) represents now the average of a total of n=4 experiments. The old graph has been replaced by the new one (see below and in the manuscript, fig 7A).

Figure 9: Effect of Ctrl or *Ptp4a2* siRNA on *Ptp4a2* and *Dll4* expression assessed by qPCR

Because Notch-1 cleavage was studied in this experiment, total Notch was now added as an internal control.

Furthermore, we have now indicated the duration of the *in vitro* DLL4 stimulation (overnight) in the Materials and Methods and Figure legend (line 409).

COMMENT 18: The changes observed in transcript level for *DLL4*, *HEY2/HES1* could be translated at the protein level? The authors should investigate the protein levels of NICD and a downstream target (*DLL4*, *HEY2* or *HES1*) in basal conditions, upon *Ptp4a2* KD.

Reply: We have studied the level of NICD released which is included in the manuscript (Figure 7b). *DLL-4*, *HEY-2* and *HES-1* expression has been determined by qPCR. Since *DLL-4*, can indeed be regulated at a translational level. An Internal Ribosome Entry Site (IRES) in the 5'-UTR of *DLL4* mRNA which is activated under hypoxic and ER stress conditions. The of the ternary complex Met-tRNAi-eIF2-GTP can regulator *DLL4* IRES-mediated translation under hypoxia¹⁶. Measuring *DLL-4* protein-levels will not give more insights how *DLL-4* expression is regulated at a translational level. *HEY-2* and *HES-1* seem not to be regulated at a translational level but protein degradation may still be an issue. Determining the protein levels of other components of the Notch pathway will not, to our opinion, add significant additional information to the mechanism underlying PRL-2 effects on vascular development.

COMMENT 19: The authors observed that *Ptp4a2* inactivation results in reduced number of retinal arteries and veins. As also suggested and discussed, PRL-2 could regulate this process through VEGF and NOTCH signaling, which have been shown to regulate arteriovenous specification (for review see Lin FJ et al., EMBO Rep, 2007; Fang JF et al., F1000Res, 2019). Furthermore, ERK signaling has been shown to regulate not only endothelial cell proliferation and cell migration but also artery formation (Srinivasan R et al., PLoS One, 2009; Lanahan A et al., Dev Cell, 2010). The authors should better connect PRL-2-VEGF/Notch-pERK-artery formation in their Discussion.

Reply: This was discussed according to the suggestions of the reviewer. We stated the following (line 308-314):

“The link between NOTCH, VEGF and PRL-2 signaling remains to be fully elucidated. We have shown that *DLL-4* ligand expression is impaired in *Ptp4a2* knockdown endothelial cells as well as NOTCH and VEGF signaling. Thus, the phenotype observed in *Ptp4a2*^{fl/fl} iEKO mice is most likely due to a combined effect on these different signaling pathways. This is consistent with the known effects of VEGF-A on *DLL-4* and notch signaling¹⁷⁻¹⁹. Furthermore, this is important because VEGF²⁰, NOTCH²¹, *DLL-4*^{22,23}, and ERK²⁴ signaling are all involved in arteriogenesis and we indeed observed a defect in the patterning of veins and arteries”

COMMENT 20: Among the multiple roles VEGF has on endothelial cells are regulation of cell migration and barrier formation. In this study, the *Ptp4a2* iEKO retinal vascular phenotype could also be attributed to a cell migration defect, due to reduced pERK1/2 or alteration of another signaling downstream of VEGFR2. Please discuss.

Reply: As it can be seen, the *in vitro* data indicate a migration defect in PRL-2 knock-down cells. This of course, is compatible with delay in vessel formation in the retina and

an alteration of the ERK1/2 pathway. It is known that ERK1/2 is involved in endothelial cell migration²⁵. This was mentioned in the manuscript (line 302)

COMMENT 21: In Fig.5C *siPtp4a2* KD cells appear more elongated than control cells at 16h post-wounding. Cell size increased? There are cell-cell adhesion defects? To evaluate cell morphology, adhesion and migration defects the authors should immunostain for junctional markers and F-actin. What represents individual data point from Fig. 5c?

Reply: According to the reviewer's comment, we have quantified cell shape change upon *Ptp4a2* siRNA treatment (see figure 11 below). No differences were detected. We included these results in a new figure (supplementary figure 5 and line 199-201).

Figure 11: Analysis of cell shapes in culture after *siPtp4a2* knock-down. Each individual point represents a cell from a biologic replicate from $n=3$ experiment.

In figure 5c (now figure 5d) Individual data points represents a biological replicate from $n=3$ independent migration experiments. This has been added in the Legend of figure 5d.

COMMENT 22: Furthermore, it is unclear how the quantitation of EdU+ ECs was done in Fig. 5d. This information is missing from the Methods or the figure legend. The authors should quantify the ratio of EdU + cells from total no. of cells in several fields in at least three independent experiments.

Reply: We provide now detailed information about the quantification in the article (line 447-454). In the figure of the previous version, a mean gray value was calculated for EdU+ quantification. To improve accuracy, the images were quantified by counting

nucleus by hand in 2-3 different biological replicates in 4 fields (500x500um) each and from n=3 independent experiment. One value was considered has an outlier.

Minor points:

COMMENT 23: The title of the manuscript should reflect the newly discovered role of PRL-2.

Reply: The title of the article was changed to: “A role of PRL-2 phosphatase in vascular morphogenesis and angiogenic signaling” .

COMMENT 24: The legends for Fig. 5 and 6 are reversed. The legends for Suppl. Fig. 5 and 6 are missing.

Reply: This has been corrected

COMMENT 25: The authors should pay attention that protein and gene names to be consistently written throughout the text, the corresponding figure legend and figure.

Reply: We verified and corrected the gene and protein names in the text.

COMMENT 26: Molecular weights should be indicated for all Western blotted proteins.

Reply: We have added the Molecular weight information on all western blots

COMMENT 27: In Fig. 2e, it would be more informative to separately display the superficial and the deeper plexus.

Reply: We did not analyze the deep plexus by separate imaging of Z-stacks, but we quantified both, WT and *Ptp4a2*^{-/-} retinas by determining the areas where formation of the deep retinal capillary plexus has been initiated and/or where the network has further expanded (blue in the figure 12 below). We normalized this area to the total area of the vascular network (yellow in the figure 12). Using this type of analysis, significant differences were seen.

Figure 12: Analysis of P9: vascularized deep plexus. (%)

COMMENT 28: In Fig. 3c and its legend and text there is a discrepancy regarding the tissue used for the analysis. Please correct.

Reply: These were Lung endothelial cells. This is now indicated in the legend. (line 732-733)

COMMENT 29 In Fig. 1c legend or in Methods part the authors should explain that the antibody used detects both PRL-1 and PRL-2 proteins. Cat. numbers for antibodies used for immunohistochemistry are missing.

Reply: We have completed the required information regarding the antibodies used. The antibody used for PRL2 detect both PRL-1 and PRL-2 (PRL-1 upper band, PRL-2 lower band). (line 197-198 in the main text and line 481 Materiel and Method)

COMMENT 30: For the scratch wound assay the text and the figure indicate 16h as the final time point, whereas in the Methods section was indicated 18h.

Reply: This was corrected (line 444). It is indeed 16h.

Reviewer #2 (Remarks to the Author)

Information to the reviewer: A detailed description of the modification of the Figures (figure modification sum-up) is added at the end of the reply to the reviewers.

COMMENT 1: The manuscript by Poulet et al. describe the first functional analysis of a PRL phosphatase, PRL-2/Ptp4a2, during vascular morphogenesis. Due to the identification of low enzymatic activity of this family of phosphatases, tissue specific function for this phosphatase *in vivo* has thus far not been studied in depth. This study shows that PRL-2 is clearly required for angiogenesis by convincing high-quality data. The authors have hereby identified an intriguing novel pathway in vascular biology. I suggest that the manuscript is acceptable for publication if minor comments below are addressed.

Reply: We are grateful for the positive assessment of our manuscript. Below is a point-by-point reply to your question and comments.

COMMENT 2: The authors show that there is an increased number of empty collagen sleeves in the EC specific KO. There is no background provided on why this is used in the field as a general read-out of vascular pruning. I suggest the authors add a sentence or two to lead into this result.

Reply: This has been done. We have written the following (line146-149):

“In order to investigate vascular stability, we labeled retinas with IB4 and an antibody recognizing the basement membrane protein collagen IV (Coll IV), and quantified IB4-negative, Coll IV-positive empty basement membrane sleeves that are left behind when vessels regress.”

COMMENT 3: Further mechanistic studies implicate a role for PRL-2 in proliferation via Notch signaling and VEGFR2 signaling. How do these pathways relate to the enhanced vascular pruning?

Reply: A potential role of PRL-2 in vascular stability and pruning may be due to altered Notch signaling regulated by PRL-2. Notch-1 was shown to be involved in vessel pruning^{10,11} and thus, PRL-2 may control stability and pruning via this mechanism. It is known that γ -secretase activity is highly sensitive magnesium¹². Since PRL-2 is involved in magnesium metabolism¹³, its effect on Notch may be indirect through the regulation of γ -secretase activity. This has not yet been clearly addressed in endothelial cells but should be part of a follow-up study.

COMMENT 4: Does the antibody used to verify PRL-2 knockdown *in vitro* (Fig5) also recognize PRL-1? Is this upregulated in the knockdown cells?

Reply: The reviewer has raised an interesting point. The antibody used in this study recognizes both PRL-1 (upper band) and PRL-2 (lower band). As suggested by the reviewer, we have performed quantification of PRL-1 protein after *Ptp4a2* knock-down in HUVECs and added these results in the manuscript (line 197-198). PRL-2 protein expression was strongly decreased and a slight increase in PRL-1 was seen in western blots after *Ptp4a2* knock-down when a large number of blots were analyzed (n=10). This has now been added to the manuscript (Figure 5 of the rebuttal letter, figure 4c of the manuscript). PRL-1 may compensate the activity of PRL-2 to some extent, however PRL-2 knock-down cells still exhibit strong phenotypic modifications and changes in cell signaling.

Figure 1: PRL-1 and PRL-2 expression quantification assessed by Western blot analysis after *Ptp4a2* knockdown (n=10)

COMMENT 5: Are there any morphologic changes to knockdown *Ptp4a2* ECs i.e are there any differences in the expression of VE-cadherin expression or FA markers that can be explained by Notch signaling deficiency and or compromised VEGF signaling?

Reply: According to the reviewer's comment, we have quantified cell shape change upon *Ptp4a2* siRNA treatment. No differences were detected. As it can be seen in the figure 2 below (or supplementary figure 5), VE-cadherin junctions are preserved.

Further experiments are needed to determine PRL2's signaling mechanisms and substrates leading to Notch and VEGFR2 signaling.

Figure 2: Analysis of cell shapes in culture after *siPtp4a2* knock-down. Each individual point represents a cell from a biologic replicate from $n=3$ experiment.

Reviewer #3 (Remarks to the Author)

Information to the reviewer: A detailed description of the modification of the Figures (figure modification sum-up) is added at the end of the reply to the reviewers.

COMMENT 1: The authors investigate the biological function of PRL phosphatases in developmental angiogenesis/morphogenesis. They focus on the specific phosphatase PRL-2, encoded by *Ptp4a2*, which they suggest is enriched in endothelial cells (ECs) of the developing retina. Global or early postnatally induced endothelial cell-specific deletion of *Ptp4a2* causes reduced/delayed vascular outgrowth with general patterning defects such as increased regression (reduced stability) and fewer arteries and veins. The authors claim that the phenotype is a consequence of alterations in the Notch pathway and that this lies downstream of PRL-2.

Although the expression data is not novel or even deeply refined this is the first focused study addressing the importance of PRL-2 in the vascular development and to my knowledge the first example of EC-specific deletion of *Ptp4a2*.

Reply: we thank the reviewer for his appreciation of our work.

COMMENT 2: *Ptp4a2* is a ubiquitously expressed gene that is slightly enriched in ECs as judged from several recently published data sets acquired through single cell transcriptomics. It is advisable to nuance the EC-selectivity and expression pattern and to direct readers to Sabbagh et al eLife 2018;7:e36187.

Reply: We agree with the reviewer and analyzed expression of the other PRLs in endothelial cells (new figure supplementary 2 and line 79-118). We also discussed the potential impact of other PRLs on the phenotypic modifications observed (line 318-323). We describe here a sum up of what we added on the manuscript. Recent works from several laboratories have provided scRNA sequencing data. (Tabula Muris: <https://tabula-muris.ds.czbiohub.org/>⁶ ; searchable database: <http://betsholtzlab.org/VascularSingleCells/database.htm>⁷ ; Trans-omics Resource Database (VECTRDB)(<https://markfsabbagh.shinyapps.io/vectrdb/>⁸. This is discussed now in detail in the manuscript and we have added a supplementary figure. From the data obtained from the Tabula Muris data base (supplementary Figure 1a and Figure 1 in the letter), we see that the expression of *Ptp4a2* is enriched in endothelial cells when compared to the other cell types in the brain (Endothelial cells: 4.13, oligodendrocytes precursor cells: 4.10, neurons: 3.93, oligodendrocytes: 3.89). In the data base from Betsholtz and collaborator, *Ptp4a2* is enriched in endothelial cells (when compared to other cell types) and *Ptp4a3* is more expressed in vSMC (supplementary Figure 1e and Figure 2 in the letter), According to the reviewer suggestions, we have also included the analysis of an additional data set from the Vascular Endothelial Cell Trans-omics Resource Database (VECTRDB) (<https://markfsabbagh.shinyapps.io/vectrdb/>⁸). Using the scRNA data set of this database, different endothelial cell types were analyzed in the brain. This latter analysis provides information on *Ptp4a2* expression in the different endothelial cell cell types (included in the manuscript (supplementary Figure 1f and Figure 3 in the letter). *Ptp4a2* is globally expressed more uniformly in all endothelial cell types.

Figure 1: Analysis of *Ptp4a1-3* expression single-cell transcriptomic data from the model organism *Mus musculus* (searchable database Tabula Muris

muris.ds.czbiohub.org/)⁶. *Ptp4a1-3* expression for each cell type sorted by FACs in brain (non-myeloid cells) can be visualized with violin plots.

Figure 2: Analysis of *Ptp4a1-3* expression in the single cell RNA sequencing dataset. (searchable database: <http://betsholtzlab.org/VascularSingleCells/database.html>)⁷ Average

expression in each cluster [Brain data] PC - Pericytes; SMC - Smooth muscle cells; MG - Microglia; FB - Vascular fibroblast-like cells; OL - Oligodendrocytes; EC - Endothelial cells; AC - Astrocytes; v - venous; capil - capillary; a - arterial; aa - arteriolar; 1,2,3- subtypes.

Figure 3: Analysis of *Ptp4a1-3* expression in Vascular Endothelial Cell Trans-omics Resource Database (VECTRDB) (<https://markfsabbagh.shinyapps.io/vectrdb/>)⁸. *Ptp4a1-3* gene expression in the developing CNS at single cell resolution. Single-cell RNA-seq on 3,946 FACS-purified GFP-positive endothelial cells from a P7 Tie2-GFP mouse brain. This dataset allows assessment of *Ptp4A1-3* expression in arterial, venous, capillary, mitotic, and tip subtypes of brain endothelial cells.

COMMENT 3: The manuscript is clearly written but contain some mismatches in main text vs figures and legends that need to be corrected. As seen below some of the claims are advised to be rephrased.

Reply: This has been corrected.

COMMENT 4: Importantly the level of experimental details must be improved, especially in terms of image quantification (see comments below).

Reply: This has been done. We also added additional information in the material and method section related to image quantification (line 356-382 of the manuscript).

COMMENT 5: The authors need to provide detailed descriptions on procedures for quantification of images (vascular density, proliferation, collagen sleeves and deep vascular plexus...). What structures are considered sleeves in contrast to only collagen IV deposits around existing vessels?

Reply: We have done all the requested modifications (Line 356-382 of the manuscript).

COMMENT 6: The word “essential” implies that no Notch signalling is possible without PRL-2. Authors show that it affects notch downstream components *in vitro* but provide no data from mice. This would be required to include such a statement.

Reply: We modified the title of the manuscript accordingly. It is now: “A role of PRL-2 phosphatase in vascular morphogenesis and angiogenic signaling”.

COMMENT 7: Abstract Row 30, Reads “Thus, our study defines PRL-2 as a major regulator of the vascular development underlying its pro-oncogenic potential reported in several advanced cancers”. However, it is a bit of a leap to infer that this would underly the cancer-phenotypes. Perhaps say that it: “may add to its pro-ocogenic...)

Reply: This has been corrected. We have removed the statement on Cancer.

COMMENT 8: Introduction Line 41: Reads “total absence of vascularisation” which is an over statement, likely more: severely defective vasculature

Reply: This has been corrected according to the reviewer’s suggestion. We exchange by: “Indeed, the loss of a single allele of the *Vegfa* gene in mice is sufficient to cause an embryonic lethality at E8.5 due to a severely defective vasculature” (line 38-39).

COMMENT 9: Line 66: Reads “dual-specificity”. It might be useful to explain for what. Results Perhaps not call the retina a “model” but rather an anatomical location/organ. Should read VEGFA not VEGFa, throughout (unless referring to the gene)

Reply: This has been explained and the other items have been corrected. This refers as specificity for both phosphotyrosine and phosphoserine (line 64).

COMMENT 10: Row 83, please clarify that this data derives from published work.

Reply: This has been done. (Line 105-107) “Online bulk RNA sequencing data from P6 to P21 mouse retinal endothelial cells revealed developmental regulation of *Ptp4a2* expression, with higher levels at early time points (P6-P15) and a decrease at P21.”

COMMENT 11: Row 84, clarify that this is whole retina, not enriched ECs.

Reply: This has been corrected (line 110)

COMMENT 12: Row 112: language issues

Reply: This has been corrected

COMMENT 13: Rows 114, 115 language

Reply: This has been corrected

COMMENT 14: Row 134, Says “strong and significant decrease”, should be sufficient to say “decreased by...” since the percentage is indicated.

Reply: This has been corrected. (line 131-133). “Analysis of P6 IsolectinB4-stained retinal whole-mounts showed an important reduction of vascular development in *Ptp4a2*^{fliEKO} retinas when compared to controls. Quantifications revealed a 40% decrease of vascular outgrowth”

COMMENT 15: Row 196, It is not the NICD “expression” that is investigated, please change. Figures Several graphs/plots do not start at 0. Please indicate that the y-axis is distorted to increase clarity (ex Fig 3c,d; 4a...).

Reply: The statement with regard to NICD has been modified. We replaced it by (line 239-240): “Upon VEGF-A stimulation, an almost complete shutdown of NICD release was observed when *Ptp4a2* was knocked-down”.

We finally choose to not distorted the Y-axis in the revised figure (and in all other figures), and replaced these figures (Figures 3c,d, Figure 4a).

COMMENT 16: It is unlikely that complete retinal material would represent EC-derived *Ptp4a2* as the ECs compose less than 1% of all cells in the retina. This limits the impact of Fig. 1 b. The data would be dramatically improved if instead ECs were isolated and analysed. Alternatively, complete material of iEC *Ptp4a2* KO retinas could be analysed to indicate the impact of EC-derived *Ptp4a2* on levels from complete retinas, as proof of principle. If this is not possible, I suggest removing this data set and/or find support in recent literature on retinal transcriptomics.

Reply: We agree with the reviewer that *Ptp4a2* levels in the whole retina of full body *Ptp4a2*^{-/-} mice do not reflect specifically endothelial cell expression. However, in silico expression on retina endothelial cells indicate that *Ptp4a2* is highly expressed which mirrors the expression on whole retinas in our study. Indeed, Jeong et al.⁹ find an increase in *Ptp4a2* expression between P6 and P15 which is in agreement with our data. These data have been now moved in supplementary data (Supplementary Fig.2).

COMMENT 17: Indicate molecular weights in blots

Reply: This has been corrected

COMMENT 18: Reads “100 ug” which is not aligned with materials and methods, please correct.

Reply: This has been corrected in the figure. 50ug was injected in postnatal mice per injection. (P0, P1, P2)

COMMENT 19: This picture is repeated in Fig 6a lower panel (mirrored), -replace image.

Reply: We thank the reviewer for spotting this and apologized for the mistake. This has been corrected.

COMMENT 20: pictures and quantification do not seem to match up. Looking in detail there seem to be Ki67 positive signal in all cells that are ERG positive which should not be the case and contradicts quantification that indicate only 10 % proliferative cells in control condition. Please clarify this and specify how the intensity comparisons were done providing image examples containing original data. How % intensity can be translated into data on proliferation would be helpful.

Reply: We assessed KI67 intensity and normalized it by ERG1/2/3 intensity. To do so, the same threshold was applied for all retina on the ki67 channel. This threshold detects cells with a bright signal and only these cells were included in the quantification since proliferating cells are KI67 high²⁶. We have put in the manuscript the original, and a modified figure is shown here in which only the cells that were counted are highlighted. The images from ERG1/2/3 channel were binarized, The ROI of ERG1/2/3+ nuclei was selected and applied to both channels (ERG1/2/3 and KI67). The intensity for this ROI (ERG1/2/3+ nuclei) was calculated for each channel, and intensity of KI67 was normalized by ERG1/2/3 intensity. For each mouse four 10X images were taken and the averaged value determined. This has been now explained in Materials and Methods (line 376-380).

Figure 4: The figure depicts the different steps for quantifying proliferation in the iEKO retinas by assessing KI67 intensity (normalized by ERG1/2/3 intensity).

COMMENT 21: How big is a field? 5. See fig legend, mixup

Reply: the field is 350x350 micron

COMMENT 22: Y-axis legend cannot be correct, likely to be number of protrusions/bead, please comment.

Reply: This has been corrected. Indeed, the number of protrusions per beads were quantified.

COMMENT 23: See fig legend mixup

Reply: This has been corrected

COMMENT 24: Composes one of the most interesting data with seemingly dramatic results. It is not completely clear how Notch cleavage is affected. To add to this, cells

could be stimulated and stained for NICD to show protein distribution. Upon VEGFA stimulation NICD is completely gone suggesting complete blockage of NOTCH signaling in the absence of the phosphatase. Such strong effect on NICD is very intriguing. An experimental repetition would be valuable to clarify solidity.

Reply: The experiment has been repeated 3 times and the data are clear and fully reproducible (figure 5 bellow). Furthermore, semi-quantitative analysis has been performed and a figure is now included (figure 7c).

Figure 5: Analysis of NOTCH-1 cleavage assessed by western blot (n=3 experiments). Quantification of NICD/NOTCH-1-TM. NICD: Notch Intra Cellular Domain, NTM: NOTCH Transmembrane.

COMMENT 25: efficiency of Notch1 siRNA-mediated knockdown is not demonstrated, please include.

Reply: We have performed a qPCR analysis on these samples which shows efficient knock-down. (figure 6).

Figure 6: Validation of Notch1 knockdown by qPCR analysis

COMMENT 26: Lacks “f”. There seem to be only two data points for DLL4 in a, no statistics can hence be applied, please comment.

Reply: We apologize for this mistake and this has been corrected in the manuscript (n=4)

Figure 7: Effe
qPCR.

ssessed by

COMMENT 27: Figure legends 1. should be “expressed” a, Should be reference 53.

b. Clarify that it is complete retinal tissue.

c. Indicate size

2. Grammar... 3. 4. 5 (should be 6). Swapped places with Fig. 6.

Reply: This has been corrected on the revised figure

COMMENT 28: Arteries and veins do form but they are fewer. More accurate to state: PRL-2 I essential for patterning of veins and arteries.

6. (should be 5)

Reply: This has been corrected according to the indications of the reviewer. The title of this figure (now Figure 4) is now (line 769): “PRL-2 is essential for arterial-venous patterning.”

COMMENT 29: Title includes data that is not shown in the figure, please correct. No panel for “f” as referred to.

Reply: The title of this figure (figure 7now) has been changed. It was: ‘PRL-2 regulates Notch1 cleavage as well as ERK1/2 downstream signaling pathways’ and we changed it to “PRL-2 regulates NOTCH-1 cleavage” (line 809).

Figure Modification Sum-Up

General:

- We have refined the statistical analysis in the figures by distinguishing between ***(p-value<0.001) and *****(p-value<0.0001)
- Panel labelling has been changed
- Molecular weight indicators were added to all western blot
- The names for genes, proteins, and genetically-engineered mice have been uniformized
- New supplementary figures have been added: supplementary figure1 and supplementary figure 5
- New supplementary figures with all uncropped WB have been added: supplementary Figure 7 and 8
- All Y- axis distortions were replaced in the revised figures (principal and supplementary figures)

Re-labeling and corrections of different figures: (first version → revised version)

Figure 1a has been removed

Figure 1b,c is now supplementary Figure 2b,c.

c. Actin in this figure was mirrored, this has been now corrected

Figure 2 is now Figure 3

Figure 3 is now Figure 1

a. tamoxifen concentration has been modified

d. pictures of Ptp4a2^{fl/fl} iEKO has been changed

e. vascular density has been re-quantified similar way than for Figure 1.f and more retinas were included

Figure 4 is now Figure 2

d. to be consistent with the other data in this figure, each individual data point represents a mouse (average of four 10X images per mouse).

Figure 5 (stays Figure 5)

a. n was increased for ptp4a2 and ptp4a3

c. PRL-1 protein quantification after *siCtrl* or *siPtp4a2* treatment is now part of this panel

f. the figure has been changed

I,j,k. Y-axis were clarified

Figure 6 is now Figure 4

Figure 7 (stay Figure 7):

a. this graph has been corrected with n=4

b. total Notch1 was added as internal control for these western blots

c. quantification of NICD cleavage is represented in a new panel

Supplementary Figure 1 is now supplementary Figure 2a

a. ptp4a3 expression has been added to this panel

Supplementary Figure 1 is now supplementary Figure 2a

Supplementary Figure 2 is now supplementary Figure 4

All graph were uniformized the same way as for the other graphs

Supplementary Figure 3 is now supplementary Figure 4:

The labelling of n was modified

Supplementary Figure 4 is now supplementary Figure 6a:

Supplementary Figure 5 is now Figure 6:

“min” was added on x-axis

Supplementary Figure 5 is now supplementary Figure 6b,c

REFERENCE

- 1 Zeng, Q., Hong, W. & Tan, Y. H. Mouse PRL-2 and PRL-3, two potentially prenylated protein tyrosine phosphatases homologous to PRL-1. *Biochem Biophys Res Commun* **244**, 421-427, doi:10.1006/bbrc.1998.8291 (1998).
- 2 McParland, V. *et al.* The metastasis-promoting phosphatase PRL-3 shows activity toward phosphoinositides. *Biochemistry* **50**, 7579-7590, doi:10.1021/bi201095z (2011).
- 3 Li, L. *et al.* Upregulation of metastasis-associated PRL-3 initiates chordoma in zebrafish. *Int J Oncol* **48**, 1541-1552, doi:10.3892/ijo.2016.3363 (2016).
- 4 Guo, K. *et al.* PRL-3 initiates tumor angiogenesis by recruiting endothelial cells in vitro and in vivo. *Cancer Res* **66**, 9625-9635, doi:10.1158/0008-5472.CAN-06-0726 (2006).
- 5 Zimmerman, M. W. *et al.* Protein-tyrosine phosphatase 4A3 (PTP4A3) promotes vascular endothelial growth factor signaling and enables endothelial cell motility. *J Biol Chem* **289**, 5904-5913, doi:10.1074/jbc.M113.480038 (2014).
- 6 Tabula Muris, C. *et al.* Single-cell transcriptomics of 20 mouse organs creates a Tabula Muris. *Nature* **562**, 367-372, doi:10.1038/s41586-018-0590-4 (2018).
- 7 Vanlandewijck, M. *et al.* A molecular atlas of cell types and zonation in the brain vasculature. *Nature* **554**, 475-480, doi:10.1038/nature25739 (2018).
- 8 Sabbagh, M. F. *et al.* Transcriptional and epigenomic landscapes of CNS and non-CNS vascular endothelial cells. *Elife* **7**, doi:10.7554/eLife.36187 (2018).
- 9 Jeong, H. W. *et al.* Transcriptional regulation of endothelial cell behavior during sprouting angiogenesis. *Nat Commun* **8**, 726, doi:10.1038/s41467-017-00738-7 (2017).
- 10 Korn, C. & Augustin, H. G. Mechanisms of Vessel Pruning and Regression. *Dev Cell* **34**, 5-17, doi:10.1016/j.devcel.2015.06.004 (2015).
- 11 Lobov, I. B. *et al.* The Dll4/Notch pathway controls postangiogenic blood vessel remodeling and regression by modulating vasoconstriction and blood flow. *Blood* **117**, 6728-6737, doi:10.1182/blood-2010-08-302067 (2011).
- 12 Ho, M. *et al.* Effect of Metal Chelators on gamma-Secretase Indicates That Calcium and Magnesium Ions Facilitate Cleavage of Alzheimer Amyloid Precursor Substrate. *Int J Alzheimers Dis* **2011**, 950932, doi:10.4061/2011/950932 (2010).
- 13 Hardy, S. *et al.* The protein tyrosine phosphatase PRL-2 interacts with the magnesium transporter CNNM3 to promote oncogenesis. *Oncogene* **34**, 986-995, doi:10.1038/onc.2014.33 (2015).
- 14 Hellstrom, M. *et al.* Dll4 signalling through Notch1 regulates formation of tip cells during angiogenesis. *Nature* **445**, 776-780, doi:10.1038/nature05571 (2007).
- 15 Pitulescu, M. E. *et al.* Dll4 and Notch signalling couples sprouting angiogenesis and artery formation. *Nat Cell Biol* **19**, 915-927, doi:10.1038/ncb3555 (2017).
- 16 Jaud, M. *et al.* The PERK Branch of the Unfolded Protein Response Promotes DLL4 Expression by Activating an Alternative Translation Mechanism. *Cancers (Basel)* **11**, doi:10.3390/cancers11020142 (2019).
- 17 Lobov, I. B. *et al.* Delta-like ligand 4 (Dll4) is induced by VEGF as a negative regulator of angiogenic sprouting. *Proc Natl Acad Sci U S A* **104**, 3219-3224, doi:10.1073/pnas.0611206104 (2007).
- 18 Liu, Z. J. *et al.* Regulation of Notch1 and Dll4 by vascular endothelial growth factor in arterial endothelial cells: implications for modulating arteriogenesis and angiogenesis. *Mol Cell Biol* **23**, 14-25, doi:10.1128/mcb.23.1.14-25.2003 (2003).

- 19 Ubezio, B. *et al.* Synchronization of endothelial Dll4-Notch dynamics switch blood vessels from branching to expansion. *Elife* **5**, doi:10.7554/eLife.12167 (2016).
- 20 Nagy, J. A. *et al.* VEGF-A induces angiogenesis, arteriogenesis, lymphangiogenesis, and vascular malformations. *Cold Spring Harb Symp Quant Biol* **67**, 227-237, doi:10.1101/sqb.2002.67.227 (2002).
- 21 Fang, J. S. *et al.* Shear-induced Notch-Cx37-p27 axis arrests endothelial cell cycle to enable arterial specification. *Nat Commun* **8**, 2149, doi:10.1038/s41467-017-01742-7 (2017).
- 22 Lobov, I. & Mikhailova, N. The Role of Dll4/Notch Signaling in Normal and Pathological Ocular Angiogenesis: Dll4 Controls Blood Vessel Sprouting and Vessel Remodeling in Normal and Pathological Conditions. *J Ophthalmol* **2018**, 3565292, doi:10.1155/2018/3565292 (2018).
- 23 Krebs, L. T. *et al.* Haploinsufficient lethality and formation of arteriovenous malformations in Notch pathway mutants. *Genes Dev* **18**, 2469-2473, doi:10.1101/gad.1239204 (2004).
- 24 Ricard, N., Zhang, J., Zhuang, Z. W. & Simons, M. Isoform-Specific Roles of ERK1 and ERK2 in Arteriogenesis. *Cells* **9**, doi:10.3390/cells9010038 (2019).
- 25 Pintucci, G. *et al.* Lack of ERK activation and cell migration in FGF-2-deficient endothelial cells. *FASEB J* **16**, 598-600, doi:10.1096/fj.01-0815fje (2002).
- 26 Miller, I. *et al.* Ki67 is a Graded Rather than a Binary Marker of Proliferation versus Quiescence. *Cell Rep* **24**, 1105-1112 e1105, doi:10.1016/j.celrep.2018.06.110 (2018).

REVIEWERS' COMMENTS:

Reviewer #1 (Remarks to the Author):

Communications Biology manuscript COMMSBIO-20-0877A
Response Reviewer 1

This manuscript is the first study which shows important roles of PRL-2, one of the three atypical protein tyrosine phosphatases, during vascular morphogenesis.

Endothelial cell (EC)-specific inactivation of Ptp4a2 gene, encoding PRL-2 protein, reduced blood vessel growth, vessel stability, arteriovenous patterning and increased sprouting and filopodia in early postnatal mouse retina. Global deletion of Ptp4a2 points in addition to a non-endothelial source of PRL-2, which has a repressive role on retinal vascular density.

In vitro assays indicate that loss of PRL-2 reduced EC migration but not their proliferation, reduced sprout lengths but increased the number of smaller and defective sprouts.

Mechanistically, this manuscript uncover the positive roles of PRL-2 on both VEGF-A and NOTCH signalling pathways. Nevertheless, more studies will be needed to investigate the clear relationship between PRL-2, NOTCH and VEGF during blood vessel angiogenesis.

The authors adequately addressed my questions and the revised manuscript is much improved. I suggest that the manuscript is acceptable for publication if remaining minor comments could be addressed:

1. The scale bars are missing in Figures 1d,g,i; 2a, c; 3; 4; 5f, h; suppl. Fig. 3a
2. If possible, please change KI67 display to a false brighter colour (red, yellow, magenta) to be more noticeable especially in printed version of the manuscript.
3. To correct:
 - lines 27-29: "Mechanistically, Ptp4a2 deletion limited endothelial sprouting by inhibiting endothelial cell migration and the VEGF DLL-4/NOTCH signaling pathway."
 - Sprouting was defective indeed, but number of sprouts was increased. One could understand that sprouting was reduced. Therefore, I suggest the authors to rephrase and make it clearer.
 - Figure 1a; line 122, 125 and 349: CDH5creERT2 -> Cdh5CreERT2
 - Figure 1a: PTP4A2fl/fl iEKO -> Ptp4a2 fl/fl iEKO
 - Figure 5g: EDU -> EdU
 - Suppl. Fig. 4 legend: (a) Mice weight -> (a) Body weight
 - line 82: of the mouse brain -> of the adult mouse brain
 - line 89: of brain endothelial cells -> of adult brain endothelial cells
 - line 124 and 350: correct tamoxifen regime administration
 - line 178: surrounding non-vascular -> surrounding non-endothelial
 - line 287: NCID -> NICD
 - line 300: decreased of -> decrease of
 - line 338: Germline transmitted Ptp4a... -> Germline transmitted Ptp4a2
 - line 734 introduce abbreviation of isolectin B4 (IB4)
 - line 771: SMA -> a-SMA
 - write invariably NOTCH-1, DLL-4 and isolectin B4 throughout the text and figure legends
 - Figure 7b: Notch1 and Notch1-TM -> NOTCH-1 and NOTCH1-TM
 - write invariably ERG1/2/3 throughout the figures

Reviewer #2 (Remarks to the Author):

The revised manuscript by de Poulet et al has addressed most of my comments. I appreciate the additional analysis of published single cell transcriptomes, which provides complementary evidence that PRL-2 is enriched in ECs and justifies further investigation of its function in retinal development. I suggest that the study in its current form is acceptable for publication, preferably with minor comments below incorporated.

Minor comments:

- Cell and junction morphology analysis in the new Sup Fig 5 are informative but I do believe that an additional IF to explore Focal Adhesions would have strengthened the data related to vessel pruning.
- In line 199 of the manuscript I suggest rephrasing "We have also investigated the cell shape and cells junction via VE-cadherin staining" into "We have also investigated EC shape and cell-cell junction morphology via VE-cadherin staining"
- I suggest that the authors add a schematic which shows how PRL-2 might interact with VEGF and Notch signalling to help the reader put all the data in context when getting to the conclusions/discussion.

Reviewer #3 (Remarks to the Author):

Authors have replied adequately to most questions and importantly introduced a more nuanced interpretation through addition of supplementary data and discussion. The revised manuscript is of good scientific quality and claims are supported by data. The connection between VEGFR2 and PRL-2 remains illusive and rather weak but deletion of PRL-2 clearly limits Notch activation in vitro. This finding is of high interest to the field although mechanistic understanding is lacking.

Minor corrections

Minor plural/singular errors to be corrected in figure legends.

Scale bars are generally missing and need to be introduced for all images.

Line 22: should read "remain"

29: To avoid "novel" in this context it may be advisable to rephrase as follows: ...PRL-2 in modulation of developmental angiogenesis.

46 "-" should be removed

91: remove "..."

Fig 1 a. This is not the final mating leading to experimental mice. Perhaps better indicate the final breeding scheme to get $lxlx;cre +/wt$.

Reviewer #1 (Remarks to the Author):

Comment 1 : This manuscript is the first study which shows important roles of PRL-2, one of the three atypical protein tyrosine phosphatases, during vascular morphogenesis.

Endothelial cell (EC)-specific inactivation of Ptp4a2 gene, encoding PRL-2 protein, reduced blood vessel growth, vessel stability, arteriovenous patterning and increased sprouting and filopodia in early postnatal mouse retina. Global deletion of Ptp4a2 points in addition to a non-endothelial source of PRL-2, which has a repressive role on retinal vascular density.

In vitro assays indicate that loss of PRL-2 reduced EC migration but not their proliferation, reduced sprout lengths but increased the number of smaller and defective sprouts.

Mechanistically, this manuscript uncover the positive roles of PRL-2 on both VEGF-A and NOTCH signalling pathways. Nevertheless, more studies will be needed to investigate the clear relationship between PRL-2, NOTCH and VEGF during blood vessel angiogenesis.

The authors adequately addressed my questions and the revised manuscript is much improved. I suggest that the manuscript is acceptable for publication if remaining minor comments could be addressed:

Reply: We would like to thank the reviewer for his comments and we will do the minor modifications he suggested.

Comment 2: The scale bars are missing in Figures 1d,g,i; 2a, c; 3; 4; 5f, h; suppl. Fig. 3a

Reply: this has been corrected

Comment 3: If possible, please change KI67 display to a false brighter colour (red, yellow, magenta) to be more noticeable especially in printed version of the manuscript.

Reply: This has been corrected (it is in red now)

Comment 4.

- lines 27-29: "Mechanistically, Ptp4a2 deletion limited endothelial sprouting by inhibiting endothelial cell migration and the VEGF DLL-4/NOTCH signaling pathway."

Sprouting was defective indeed, but number of sprouts was increased. One could understand that sprouting was reduced. Therefore, I suggest the authors to rephrase and make it clearer.

Reply: This has been corrected. It is stated now:

"Mechanistically, PTP4A2 deletion limits angiogenesis by inhibiting endothelial cell migration and the VEGF-A, DLL-4/NOTCH-1 signaling pathway".

Comment 5

- Figure 1a; line 122, 125 and 349: CDH5creERT2 ->Cdh5CreERT2

- Figure 1a: PTP4A2fl/fl iEKO -> Ptp4a2 fl/fl iEKO

- Figure 5g: EDU -> EdU
 - Suppl. Fig. 4 legend: (a) Mice weight-> (a) Body weight
 - line 82: of the mouse brain ->: of the adult mouse brain
 - line 89: of brain endothelial cells -> of adult brain endothelial cells
 - line 124 and 350: correct tamoxifen regime administration
 - line 178: surrounding non-vascular -> surrounding non-endothelial
 - line 287: NCID -> NICD
 - line 300: decreased of -> decrease of
 - line 338: Germline transmitted Ptp4a... ->Germline transmitted Ptp4a2
 - line 734 introduce abbreviation of isolectin B4 (IB4)
 - line 771: SMA ->a-SMA
 - write invariably NOTCH-1, DLL-4 and isolectin B4 throughout the text and figure legends
 - Figure 7b: Notch1 and Notch1-TM -> NOTCH-1 and NOTCH1-TM
 - write invariably ERG1/2/3 throughout the figures
- Reply:** all this has been corrected

Reviewer #2 (Remarks to the Author):

Comment 1: The revised manuscript by de Poulet et al has addressed most of my comments.

I appreciate the additional analysis of published single cell transcriptomes, which provides complementary evidence that PRL-2 is enriched in ECs and justifies further investigation of its function in retinal development. I suggest that the study in its current form is acceptable for publication, preferably with minor comments below incorporated.

Reply: We would like to thank the reviewer for his appreciation of our revisions

Comment 2:

- Cell and junction morphology analysis in the new Sup Fig 5 are informative but I do believe that an additional IF to explore Focal Adhesions would have strengthened the data related to vessel pruning.

Reply: It would be indeed interesting to do this experiment, however we believe that the VE-CADHERIN staining is sufficiently informative for addressing this issue.

Comment 3:

- In line 199 of the manuscript I suggest rephrasing “We have also investigated the cell shape and cells junction via VE-cadherin staining” into “We have also investigated EC shape and cell-cell junction morphology via VE-cadherin staining”

Reply: This has been corrected according to the indications of the reviewer

Comment 4- I suggest that the authors add a schematic which shows how PRL-2 might interact with VEGF and Notch signalling to help the reader put all the data in context when getting to the conclusions/discussion.

Reply: We have added a figure according to the reviewer’s suggestion at the end of the manuscript (Figure 7g).

Reviewer #3 (Remarks to the Author):

Comment 1: Authors have replied adequately to most questions and importantly introduced a more nuanced interpretation through addition of supplementary data and discussion. The revised manuscript is of good scientific quality and claims are supported by data. The connection between VEGFR2 and PRL-2 remains illusive and rather weak but deletion of PRL-2 clearly limits Notch activation in vitro. This finding is of high interest to the field although mechanistic understanding is lacking.

Reply: We would like to thank the reviewer for the supportive comments of our revised article

Comment 2:

Minor plural/singular errors to be corrected in figure legends.

Scale bars are generally missing and need to be introduced for all images.

Line 22: should read "remain"

29: To avoid "novel" in this context it may be advisable to rephrase as follows:

...PRL-2 in modulation of developmental angiogenesis.

46 "-" should be removed

91: remove "..."

Fig 1 a. This is not the final mating leading to experimental mice. Perhaps better indicate the final breeding scheme to get $lxlx;cre +/wt$.

Reply: We have corrected this according to the reviewer's comments